# The asymptotic spectrum of the Hessian of DNN throughout training

**Arthur Jacot, Franck Gabriel & Clément Hongler**
Chair of Statistical Field Theory
Ecole Polytechnique Fédérale de Lausanne
{arthur.jacot,franck.grabriel,clement.hongler}@epfl.ch

## Abstract

The dynamics of DNNs during gradient descent is described by the so-called Neural Tangent Kernel (NTK). In this article, we show that the NTK allows one to gain precise insight into the Hessian of the cost of DNNs. When the NTK is fixed during training, we obtain a full characterization of the asymptotics of the spectrum of the Hessian, at initialization and during training. In the so-called mean-field limit, where the NTK is not fixed during training, we describe the first two moments of the Hessian at initialization.

## 1 Introduction

The advent of deep learning has sparked a lot of interest in the loss surface of deep neural networks (DNN), and in particular its Hessian. However to our knowledge, there is still no theoretical description of the spectrum of the Hessian. Nevertheless a number of phenomena have been observed numerically.

The loss surface of neural networks has been compared to the energy landscape of different physical models (Choromanska et al., 2015; Geiger et al., 2018; Mei et al., 2018). It appears that the loss surface of DNNs may change significantly depending on the width of the network (the number of neurons in the hidden layer), motivating the distinction between the under- and over-parametrized regimes (Baity-Jesi et al., 2018; Geiger et al., 2018; 2019).

The non-convexity of the loss function implies the existence of a very large number of saddle points, which could slow down training. In particular, in (Pascanu et al., 2014; Dauphin et al., 2014), a relation between the rank of saddle points (the number of negative eigenvalues of the Hessian) and their loss has been observed.

For overparametrized DNNs, a possibly more important phenomenon is the large number of flat directions (Baity-Jesi et al., 2018). The existence of these flat minima is conjectured to be related to the generalization of DNNs and may depend on the training procedure (Hochreiter & Schmidhuber, 1997; Chaudhari et al., 2016; Wu et al., 2017).

In (Jacot et al., 2018) it has been shown, using a functional approach, that in the infinite-width limit, DNNs behave like kernel methods with respect to the so-called Neural Tangent Kernel, which is determined by the architecture of the network. This leads to convergence guarantees for DNNs (Jacot et al., 2018; Du et al., 2019; Allen-Zhu et al., 2018; Huang & Yau, 2019) and strengthens the connections between neural networks and kernel methods (Neal, 1996; Cho & Saul, 2009; Lee et al., 2018).

Our approach also allows one to probe the so-called mean-field/active limit (studied in (Rotskoff & Vanden-Eijnden, 2018; Chizat & Bach, 2018a; Mei et al., 2018) for shallow networks), where the NTK varies during training.

This raises the question: can we use these new results to gain insight into the behavior of the Hessian of the loss of DNNs, at least in the small region explored by the parameters during training?

## 1.1 Contributions

Following ideas introduced in (Jacot et al., 2018), we consider the training of $L+1$-layered DNNs in a functional setting. For a functional cost $\mathcal{C}$, the Hessian of the loss $\mathbb{R}^P \ni \theta \mapsto \mathcal{C}\left(F^{(L)}(\theta)\right)$ is the sum of two $P \times P$ matrices $I$ and $S$. We show the following results for large $P$ and for a fixed number of datapoints $N$:

- The first matrix $I$ is positive semi-definite and its eigenvalues are given by the (weighted) kernel PCA of the dataset with respect to the NTK. The dominating eigenvalues are the principal components of the data followed by a high number of small eigenvalues. The "flat directions" are spanned by the small eigenvalues and the null-space (of dimension at least $P - N$ when there is a single output). Because the NTK is asymptotically constant (Jacot et al., 2018), these results apply at initialization, during training and at convergence.

- The second matrix $S$ can be viewed as residual contribution to $H$, since it vanishes as the network converges to a global minimum. We compute the limit of the first moment $\text{Tr}(S)$ and characterize its evolution during training, of the second moment $\text{Tr}(S^2)$ which stays constant during training, and show that the higher moments vanish.

- Regarding the sum $H = I + S$, we show that the matrices $I$ and $S$ are asymptotically orthogonal to each other at initialization and during training. In particular, the moments of the matrices $I$ and $S$ add up: $tr(H^k) \approx tr(I^k) + tr(S^k)$.

These results give, for any depth and a fairly general non-linearity, a complete description of the spectrum of the Hessian in terms of the NTK at initialization and throughout training. Our theoretical results are consistent with a number of observations about the Hessian (Hochreiter & Schmidhuber, 1997; Pascanu et al., 2014; Dauphin et al., 2014; Chaudhari et al., 2016; Wu et al., 2017; Pennington & Bahri, 2017; Geiger et al., 2018), and sheds a new light on them.

## 1.2 Related works

The Hessian of the loss has been studied through the decomposition $I + S$ in a number of previous works (Sagun et al., 2017; Pennington & Bahri, 2017; Geiger et al., 2018).

For least-squares and cross-entropy costs, the first matrix $I$ is equal to the Fisher matrix (Wagenaar, 1998; Pascanu & Bengio, 2013), whose moments have been described for shallow networks in (Pennington & Worah, 2018). For deep networks, the first two moments and the operator norm of the Fisher matrix for a least squares loss were computed at initialization in (Karakida et al., 2018) conditionally on a certain independence assumption; our method does not require such assumptions. Note that their approach implicitly uses the NTK.

The second matrix $S$ has been studied in (Pennington & Bahri, 2017; Geiger et al., 2018) for shallow networks, conditionally on a number of assumptions. Note that in the setting of (Pennington & Bahri, 2017), the matrices $I$ and $S$ are assumed to be freely independent, which allows them to study the spectrum of the Hessian; in our setting, we show that the two matrices $I$ and $S$ are asymptotically orthogonal to each other.

## 2 Setup

We consider deep fully connected artificial neural networks (DNNs) using the setup and NTK parametrization of (Jacot et al., 2018), taking an arbitrary nonlinearity $\sigma \in C_b^4(\mathbb{R})$ (i.e. $\sigma : \mathbb{R} \to \mathbb{R}$ that is 4 times continuously differentiable function with all four derivatives bounded). The layers are numbered from 0 (input) to $L$ (output), each containing $n_\ell$ neurons for $\ell = 0, \ldots, L$. The $P = \sum_{\ell=0}^{L-1} (n_\ell + 1) n_{\ell+1}$ parameters consist of the weight matrices $W^{(\ell)} \in \mathbb{R}^{n_{\ell+1} \times n_\ell}$ and bias vectors $b^{(\ell)} \in \mathbb{R}^{n_{\ell+1}}$ for $\ell = 0, \ldots, L-1$. We aggregate the parameters into the vector $\theta \in \mathbb{R}^P$.

The activations and pre-activations of the layers are defined recursively for an input $x \in \mathbb{R}^{n_0}$, setting $\alpha^{(0)}(x; \theta) = x$ :

$$\tilde{\alpha}^{(\ell+1)}(x; \theta) = \frac{1}{\sqrt{n_\ell}} W^{(\ell)} \alpha^{(\ell)}(x; \theta) + \beta b^{(\ell)},$$

$$\alpha^{(\ell+1)}(x; \theta) = \sigma\big(\tilde{\alpha}^{(\ell+1)}(x; \theta)\big).$$

The parameter $\beta$ is added to tune the influence of the bias on training[1]. All parameters are initialized as iid $\mathcal{N}(0,1)$ Gaussians.

We will in particular study the network function, which maps inputs $x$ to the activation of the output layer (before the last non-linearity):

$$f_\theta(x) = \tilde{\alpha}^{(L)}(x; \theta).$$

In this paper, we will study the limit of various objects as $n_1, \ldots, n_L \to \infty$ *sequentially*, i.e. we first take $n_1 \to \infty$, then $n_2 \to \infty$, etc. This greatly simplifies the proofs, but they could in principle be extended to the simultaneous limit, i.e. when $n_1 = ... = n_{L-1} \to \infty$. All our numerical experiments are done with 'rectangular' networks (with $n_1 = ... = n_{L-1}$) and match closely the predictions for the sequential limit.

In the limit we study in this paper, the NTK is asymptotically fixed, as in (Jacot et al., 2018; Allen-Zhu et al., 2018; Du et al., 2019; Arora et al., 2019; Huang & Yau, 2019). By rescaling the outputs of DNNs as the width increases, one can reach another limit where the NTK is not fixed (Chizat & Bach, 2018a;b; Rotskoff & Vanden-Eijnden, 2018; Mei et al., 2019). Some of our results can be extended to this setting, but only at initialization (see Section 3.3). The behavior during training becomes however much more complex.

## 2.1 Functional viewpoint

The network function lives in a function space $f_\theta \in \mathcal{F} := [\mathbb{R}^{n_0} \to \mathbb{R}^{n_L}]$ and we call the function $F^{(L)} : \mathbb{R}^P \to \mathcal{F}$ that maps the parameters $\theta$ to the network function $f_\theta$ the *realization function*. We study the differential behavior of $F^{(L)}$:

- The derivative $\mathcal{D}F^{(L)} \in \mathbb{R}^P \otimes \mathcal{F}$ is a function-valued vector of dimension $P$. The $p$-th entry $\mathcal{D}_p F^{(L)} = \partial_{\theta_p} f_\theta \in \mathcal{F}$ represents how modifying the parameter $\theta_p$ modifies the function $f_\theta$ in the space $\mathcal{F}$.
- The Hessian $\mathcal{H}F^{(L)} \in \mathbb{R}^P \otimes \mathbb{R}^P \otimes \mathcal{F}$ is a function-valued $P \times P$ matrix.

The network is trained with respect to the cost functional:

$$\mathcal{C}(f) = \frac{1}{N} \sum_{i=1}^N c_i\left(f(x_i)\right),$$

for strictly convex $c_i$, summing over a finite dataset $x_1, \ldots, x_N \in \mathbb{R}^{n_0}$ of size $N$. The parameters are then trained with gradient descent on the composition $\mathcal{C} \circ F^{(L)}$, which defines the usual loss surface of neural networks.

In this setting, we define the finite realization function $Y^{(L)} : \mathbb{R}^P \to \mathbb{R}^{Nn_L}$ mapping parameters $\theta$ to be the restriction of the network function $f_\theta$ to the training set $y_{ik} = f_{\theta,k}(x_i)$. The Jacobian $\mathcal{D}Y^{(L)}$ is hence an $Nn_L \times P$ matrix and its Hessian $\mathcal{H}Y^{(L)}$ is a $P \times P \times Nn_L$ tensor. Defining the restricted cost $C(y) = \frac{1}{N} \sum_i c_i(y_i)$, we have $\mathcal{C} \circ F^{(L)} = C \circ Y^{(L)}$.

For our analysis, we require that the gradient norm $\|\mathcal{D}C\|$ does not explode during training. The following condition is sufficient:

**Definition 1.** A loss $C : \mathbb{R}^{Nn_L} \to \mathbb{R}$ has bounded gradients over sublevel sets (BGOSS) if the norm of the gradient is bounded over all sets $U_a = \left\{ Y \in \mathbb{R}^{Nn_L} : C(Y) \le a \right\}$.

For example, the Mean Square Error (MSE) $C(Y) = \frac{1}{2N} \|Y^* - Y\|^2$ for the labels $Y^* \in \mathbb{R}^{Nn_L}$ has BGOSS because $\|\nabla C(Y)\|^2 = \frac{1}{N} \|Y^* - Y\|^2 = 2C(Y)$. For the binary and softmax cross-entropy the gradient is uniformly bounded, see Proposition 2 in Appendix A.

---

[1] In our experiments, we take $\beta = 0.1$.

## 2.2 Neural Tangent Kernel

The behavior during training of the network function $f_\theta$ in the function space $\mathcal{F}$ is described by a (multi-dimensional) kernel, the *Neural Tangent Kernel* (NTK)

$$\Theta_{k,k'}^{(L)}(x,x') = \sum_{p=1}^{P} \partial_{\theta_p} f_{\theta,k}(x) \partial_{\theta_p} f_{\theta,k'}(x').$$

During training, the function $f_\theta$ follows the so-called *kernel gradient descent* with respect to the NTK, which is defined as

$$\partial_t f_{\theta(t)}(x) = -\nabla_{\Theta^{(L)}} C_{|f_{\theta(t)}}(x) := -\frac{1}{N} \sum_{i=1}^{N} \Theta^{(L)}(x,x_i) \nabla c_i(f_{\theta(t)}(x_i)).$$

In the infinite-width limit (letting $n_1 \to \infty, \dots, n_{L-1} \to \infty$ sequentially) and for losses with BGOSS, the NTK converges to a deterministic limit $\Theta^{(L)} \to \Theta_\infty^{(L)} \otimes Id_{n_L}$, which is constant during training, uniformly on finite time intervals $[0, T]$ (Jacot et al., 2018). For the MSE loss, the uniform convergence of the NTK was proven for $T = \infty$ in (Arora et al., 2019).

The limiting NTK $\Theta_\infty^{(L)} : \mathbb{R}^{n_0} \times \mathbb{R}^{n_0} \to \mathbb{R}$ is constructed as follows:

1. For $f, g : \mathbb{R} \to \mathbb{R}$ and a kernel $K : \mathbb{R}^{n_0} \times \mathbb{R}^{n_0} \to \mathbb{R}$, define the kernel $\mathbb{L}_K^{f,g} : \mathbb{R}^{n_0} \times \mathbb{R}^{n_0} \to \mathbb{R}$ by
   $$\mathbb{L}_K^{f,g}(x_0,x_1) = \mathbb{E}_{(a_0,a_1)} \left[ f(a_0) g(a_1) \right],$$
   for $(a_0, a_1)$ a centered Gaussian vector with covariance matrix $(K(x_i, x_j))_{i,j=0,1}$. For $f = g$, we denote by $\mathbb{L}_K^f$ the kernel $\mathbb{L}_K^{f,f}$.

2. We define the kernels $\Sigma_\infty^{(\ell)}$ for each layer of the network, starting with $\Sigma_\infty^{(1)}(x_0, x_1) = 1/n_0(x_0^T x_1) + \beta^2$ and then recursively by $\Sigma_\infty^{(\ell+1)} = \mathbb{L}_{\Sigma_\infty^{(\ell)}}^\sigma + \beta^2$, for $\ell = 1, \dots, L-1$, where $\sigma$ is the network non-linearity.

3. The limiting NTK $\Theta_\infty^{(L)}$ is defined in terms of the kernels $\Sigma_\infty^{(\ell)}$ and the kernels $\dot{\Sigma}_\infty^{(\ell)} = \mathbb{L}_{\Sigma_\infty^{(\ell-1)}}^{\dot{\sigma}}$:
   $$\Theta_\infty^{(L)} = \sum_{\ell=1}^{L} \Sigma_\infty^{(\ell)} \dot{\Sigma}_\infty^{(\ell+1)} \dots \dot{\Sigma}_\infty^{(L)}.$$

The NTK leads to convergence guarantees for DNNs in the infinite-width limit, and connect their generalization to that of kernel methods (Jacot et al., 2018; Arora et al., 2019).

## 2.3 Gram Matrices

For a finite dataset $x_1, \dots, x_N \in \mathbb{R}^{n_0}$ and a fixed depth $L \geq 1$, we denote by $\tilde{\Theta} \in \mathbb{R}^{Nn_L \times Nn_L}$ the Gram matrix of $x_1, \dots, x_N$ with respect to the limiting NTK, defined by
$$\tilde{\Theta}_{ik,jm} = \Theta_\infty^{(L)}(x_i, x_j) \delta_{km}.$$

It is block diagonal because different outputs $k \neq m$ are asymptotically uncorrelated.

Similarly, for any (scalar) kernel $\mathcal{K}^{(L)}$ (such as the limiting kernels $\Sigma_\infty^{(L)}, \Lambda_\infty^{(L)}, \Upsilon_\infty^{(L)}, \Phi_\infty^{(L)}, \Xi_\infty^{(L)}$ introduced later), we denote the Gram matrix of the datapoints by $\tilde{\mathcal{K}}$.

## 3 Main Theorems

### 3.1 Hessian as $I + S$

Using the above setup, the Hessian $H$ of the loss $\mathcal{C} \circ F^{(L)}$ is the sum of two terms, with the entry $H_{p,p'}$ given by
$$H_{p,p'} = \mathcal{HC}_{|f_\theta}(\partial_{\theta_p} F, \partial_{\theta_{p'}} F) + \mathcal{DC}_{|f_\theta}(\partial_{\theta_p,\theta_{p'}} F).$$

For a finite dataset, the Hessian matrix $\mathcal{H}\left(C \circ Y^{(L)}\right)$ is equal to the sum of two matrices

$$I = \left(\mathcal{D}Y^{(L)}\right)^T \mathcal{H}C\mathcal{D}Y^{(L)} \quad \text{and} \quad S = \nabla C \cdot \mathcal{H}Y^{(L)}$$

where $\mathcal{D}Y^{(L)}$ is a $Nn_L \times P$ matrix, $\mathcal{H}C$ is a $Nn_L \times Nn_L$ matrix and $\mathcal{H}Y^{(L)}$ is a $P \times P \times Nn_L$ tensor to which we apply a scalar product (denoted by $\cdot$) in its last dimension with the $Nn_L$ vector $\nabla C$ to obtain a $P \times P$ matrix.

Our main contribution is the following theorem, which describes the limiting moments $\text{Tr}\left(H^k\right)$ in terms of the moments of $I$ and $S$:

**Theorem 1.** *For any loss $C$ with BGOSS and $\sigma \in C_b^4(\mathbb{R})$, in the sequential limit $n_1 \to \infty, \dots, n_{L-1} \to \infty$, we have for all $k \geq 1$*

$$\text{Tr}\left(H\left(t\right)^k\right) \approx \text{Tr}\left(I\left(t\right)^k\right) + \text{Tr}\left(S\left(t\right)^k\right).$$

*The limits of $\text{Tr}\left(I\left(t\right)^k\right)$ and $\text{Tr}\left(S\left(t\right)^k\right)$ can be expressed in terms of the NTK $\Theta_\infty^{(L)}$, the kernels $\Upsilon_\infty^{(L)}, \Xi_\infty^{(L)}$ and the non-symmetric kernels $\Phi_\infty^{(L)}, \Lambda_\infty^{(L)}$ defined in Appendix C:*

- *The moments $\text{Tr}\left(I\left(t\right)^k\right)$ converge to the following limits (with the convention that $i_{k+1} = i_1$):*

$$\text{Tr}\left(I\left(t\right)^k\right) \to \text{Tr}\left(\left(\mathcal{H}C(Y\left(t\right))\tilde{\Theta}\right)^k\right) = \frac{1}{N^k}\sum_{i_1,\dots,i_k=1}^{N}\prod_{m=1}^{k} c_{i_m}''(f_{\theta(t)}(x_{i_m}))\Theta_\infty^{(L)}(x_{i_m}, x_{i_{m+1}}).$$

- *The first moment $\text{Tr}\left(S\left(t\right)\right)$ converges to the limit:*

$$\text{Tr}\left(S\left(t\right)\right) = (G(t))^T \nabla C(Y(t)).$$

*At initialization $(G(0), Y(0))$ form a Gaussian pair of $Nn_L$-vectors, independent for differing output indices $k = 1, \dots, n_L$ and with covariance $\mathbb{E}\left[G_{ik}(0)G_{i'k'}(0)\right] = \delta_{kk'}\Xi_\infty^{(L)}(x_i, x_{i'})$ and $\mathbb{E}\left[G_{ik}(0)Y_{i'k'}(0)\right] = \delta_{kk'}\Phi_\infty^{(L)}(x_i, x_{i'})$ for the limiting kernel $\Xi_\infty^{(L)}(x, y)$ and non-symmetric kernel $\Phi_\infty^{(L)}(x, y)$. During training, both vectors follow the differential equations*

$$\partial_t G(t) = -\tilde{\Lambda}\nabla C(Y(t))$$
$$\partial_t Y(t) = -\tilde{\Theta}\nabla C(Y(t)).$$

- *The second moment $\text{Tr}\left(S\left(t\right)^2\right)$ converges to the following limit defined in terms of the Gram matrix $\tilde{\Upsilon}$:*

$$\text{Tr}\left(S^2\right) \to (\nabla C(Y(t)))^T \tilde{\Upsilon}\nabla C(Y(t))$$

- *The higher moments $\text{Tr}\left(S\left(t\right)^k\right)$ for $k \geq 3$ vanish.*

*Proof.* The moments of $I$ and $S$ can be studied separately because the moments of their sum is asymptotically equal to the sum of their moments by Proposition 5 below. The limiting moments of $I$ and $S$ are respectively described by Propositions 1 and 4 below. $\qquad\square$

In the case of a MSE loss $C(Y) = \frac{1}{2N}\|Y - Y^*\|^2$, the first and second derivatives take simple forms $\nabla C(Y) = \frac{1}{N}(Y - Y^*)$ and $\mathcal{H}C(Y) = \frac{1}{N}Id_{Nn_L}$ and the differential equations can be solved to obtain more explicit formulae:

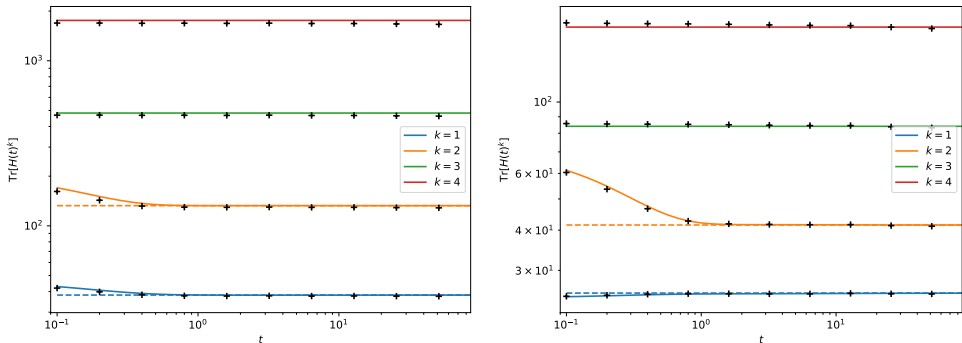

Figure 1: Comparison of the theoretical prediction of Corollary 1 for the expectation of the first 4 moments (colored lines) to the empirical average over 250 trials (black crosses) for a rectangular network with two hidden layers of finite widths $n_1 = n_2 = 5000$ ($L = 3$) with the smooth ReLU (left) and the normalized smooth ReLU (right), for the MSE loss on scaled down 14x14 MNIST with $N = 256$. Only the first two moments are affected by $S$ at the beginning of training.

**Corollary 1.** *For the MSE loss $C$ and $\sigma \in C_b^4(\mathbb{R})$, in the limit $n_1, ..., n_{L-1} \to \infty$, we have uniformly over $[0, T]$*

$$\mathrm{Tr}\left(H(t)^k\right) \to \frac{1}{N^k}\mathrm{Tr}\left(\tilde{\Theta}^k\right) + \mathrm{Tr}\left(S(t)^k\right)$$

*where*

$$\mathrm{Tr}\left(S(t)\right) \to -\frac{1}{N}(Y^* - Y(0))^T\left(Id_{Nn_L} + e^{-t\tilde{\Theta}}\right)\tilde{\Theta}^{-1}\tilde{\Lambda}^T e^{-t\tilde{\Theta}}(Y^* - Y(0))$$
$$+ \frac{1}{N}G(0)^T e^{-t\tilde{\Theta}}(Y^* - Y(0))$$
$$\mathrm{Tr}\left(S(t)^2\right) \to \frac{1}{N^2}(Y^* - Y(0))^T e^{-t\tilde{\Theta}}\tilde{\Upsilon}e^{-t\tilde{\Theta}}(Y^* - Y(0))$$
$$\mathrm{Tr}\left(S(t)^k\right) \to 0 \quad \text{when } k > 2.$$

*In expectation we have:*

$$\mathbb{E}\left[\mathrm{Tr}\left(S(t)\right)\right] \to -\frac{1}{N}Tr\left(\left(Id_{Nn_L} + e^{-t\tilde{\Theta}}\right)\tilde{\Theta}^{-1}\tilde{\Lambda}^T e^{-t\tilde{\Theta}}\left(\tilde{\Sigma} + Y^*Y^{*T}\right)\right) + \frac{1}{N}Tr\left(e^{-t\tilde{\Theta}}\tilde{\Phi}^T\right)$$
$$\mathbb{E}\left[\mathrm{Tr}\left(S(t)^2\right)\right] \to \frac{1}{N^2}Tr\left(e^{-t\tilde{\Theta}}\tilde{\Upsilon}e^{-t\tilde{\Theta}}\left(\tilde{\Sigma} + Y^*Y^{*T}\right)\right).$$

*Proof.* The moments of $I$ are constant because $\mathcal{H}C = \frac{1}{N}Id_{Nn_L}$ is constant. For the moments of $S$, we first solve the differential equation for $Y(t)$:

$$Y(t) = Y^* - e^{-t\tilde{\Theta}}(Y^* - Y(0)).$$

Noting $Y(t) - Y(0) = -\tilde{\Theta}\int_0^t \nabla C(s)ds$, we have

$$G(t) = G(0) - \tilde{\Lambda}\int_0^t \nabla C(s)ds$$
$$= G(0) + \tilde{\Lambda}\tilde{\Theta}^{-1}(Y(t) - Y(0))$$
$$= G(0) + \tilde{\Lambda}\tilde{\Theta}^{-1}\left(Id_{Nn_L} + e^{-t\tilde{\Theta}}\right)(Y^* - Y(0))$$

The expectation of the first moment of $S$ then follows. $\square$

## 3.2 Mutual Orthogonality of $I$ and $S$

A first key ingredient to prove Theorem 1 is the asymptotic mutual orthogonality of the matrices $I$ and $S$

**Proposition** (Proposition 5 in Appendix D). *For any loss $C$ with BGOSS and $\sigma \in C_b^4(\mathbb{R})$, we have uniformly over $[0, T]$*

$$\lim_{n_{L-1} \to \infty} \cdots \lim_{n_1 \to \infty} \|IS\|_F = 0.$$

*As a consequence* $\lim_{n_{L-1} \to \infty} \cdots \lim_{n_1 \to \infty} \mathrm{Tr}\left([I + S]^k\right) - \left[\mathrm{Tr}\left(I^k\right) + \mathrm{Tr}\left(S^k\right)\right] = 0.$

*Remark* 1. If two matrices $A$ and $B$ are mutualy orthogonal (i.e. $AB = 0$) the range of $A$ is contained in the nullspace of $B$ and vice versa. The non-zero eigenvalues of the sum $A + B$ are therefore given by the union of the non-zero eigenvalues of $A$ and $B$. Furthermore the moments of $A$ and $B$ add up: $\mathrm{Tr}\left([A + B]^k\right) = \mathrm{Tr}\left(A^k\right) + \mathrm{Tr}\left(B^k\right)$. Proposition 5 shows that this is what happens asymptotically for $I$ and $S$.

Note that both matrices $I$ and $S$ have large nullspaces: indeed assuming a constant width $w = n_1 = ... = n_{L-1}$, we have $Rank(I) \leq Nn_L$ and $Rank(S) \leq 2(L-1)wNn_L$ (see Appendix C), while the number of parameters $P$ scales as $w^2$ (when $L > 2$).

Figure 2 illustrates the mutual orthogonality of $I$ and $S$. All numerical experiments are done for rectangular networks (when the width of the hidden layers are equal) and agree well with our predictions obtained in the sequential limit.

## 3.3 Mean-field Limit

For a rectangular network with width $w$, if the output of the network is divided by $\sqrt{w}$ and the learning rate is multiplied by $w$ (to keep similar dynamics at initialization), the training dynamics changes and the NTK varies during training when $w$ goes to infinity. The new parametrization of the output changes the scaling of the two matrices:

$$\mathcal{H}\left[C\left(\frac{1}{\sqrt{w}}Y^{(L)}\right)\right] = \frac{1}{w}\left(\mathcal{D}Y^{(L)}\right)^T \mathcal{H}C\mathcal{D}Y^{(L)} + \frac{1}{\sqrt{w}}\nabla C \cdot \mathcal{H}Y^{(L)} = \frac{1}{w}I + \frac{1}{\sqrt{w}}S.$$

The scaling of the learning rate essentially multiplies the whole Hessian by $w$. In this setting, the matrix $I$ is left unchanged while the matrix $S$ is multiplied by $\sqrt{w}$ (the $k$-th moment of $S$ is hence multiplied by $w^{k/2}$). In particular, the two moments of the Hessian are dominated by the moments of $S$, and the higher moments of $S$ (and the operator norm of $S$) should not vanish. This suggests that the active regime may be characterised by the fact that $\|S\|_F \gg \|I\|_F$. Under the conjecture that Theorem 1 holds for the infinite-width limit of rectangular networks, the asymptotic of the two first moments of $H$ is given by:

$$^1/\sqrt{w}\mathrm{Tr}\left(H\right) \to \mathcal{N}(0, \nabla C^T\tilde{\Xi}\nabla C)$$
$$^1/w\mathrm{Tr}\left(H^2\right) \to \nabla C^T\tilde{\Upsilon}\nabla C,$$

where for the MSE loss we have $\nabla C = -Y^*$.

## 3.4 The matrix $S$

The matrix $S = \nabla C \cdot \mathcal{H}Y^{(L)}$ is best understood as a perturbation to $I$, which vanishes as the network converges because $\nabla C \to 0$. To calculate its moments, we note that

$$\mathrm{Tr}\left(\nabla C \cdot \mathcal{H}Y^{(L)}\right) = \left(\sum_{p=1}^{P}\partial_{\theta_p^2}^2 Y\right)^T \nabla C = G^T \nabla C,$$

where the vector $G = \sum_{k=1}^{P}\partial_{\theta_p^2}^2 Y \in \mathbb{R}^{Nn_L}$ is the evaluation of the function $g_\theta(x) = \sum_{k=1}^{P}\partial_{\theta_p^2}^2 f_\theta(x)$ on the training set.

For the second moment we have

$$\mathrm{Tr}\left(\left(\nabla C \cdot \mathcal{H}Y^{(L)}\right)^2\right) = \nabla C^T \left(\sum_{p,p'=1}^{P} \partial^2_{\theta_p \theta_{p'}} Y \left(\partial^2_{\theta_p \theta_{p'}} Y\right)^T\right) \nabla C = \nabla C^T \tilde{\Upsilon} \nabla C$$

for $\tilde{\Upsilon}$ the Gram matrix of the kernel $\Upsilon^{(L)}(x,y) = \sum_{p,p'=1}^{P} \partial^2_{\theta_p \theta_{p'}} f_\theta(x) \left(\partial^2_{\theta_p \theta_{p'}} f_\theta(y)\right)^T$.

The following proposition desribes the limit of the function $g_\theta$ and the kernel $\Upsilon^{(L)}$ and the vanishing of the higher moments:

**Proposition** (Proposition 4 in Appendix C). *For any loss $C$ with BGOSS and $\sigma \in C_b^4(\mathbb{R})$, the first two moments of $S$ take the form*

$$\mathrm{Tr}\left(S(t)\right) = G(t)^T \nabla C(t)$$
$$\mathrm{Tr}\left(S(t)^2\right) = \nabla C(t)^T \tilde{\Upsilon}(t) \nabla C(t)$$

*- At initialization, $g_\theta$ and $f_\theta$ converge to a (centered) Gaussian pair with covariances*

$$\mathbb{E}[g_{\theta,k}(x)g_{\theta,k'}(x')] = \delta_{kk'} \Xi_\infty^{(L)}(x,x')$$
$$\mathbb{E}[g_{\theta,k}(x)f_{\theta,k'}(x')] = \delta_{kk'} \Phi_\infty^{(L)}(x,x')$$
$$\mathbb{E}[f_{\theta,k}(x)f_{\theta,k'}(x')] = \delta_{kk'} \Sigma_\infty^{(L)}(x,x')$$

*and during training $g_\theta$ evolves according to*

$$\partial_t g_{\theta,k}(x) = \sum_{i=1}^{N} \Lambda_\infty^{(L)}(x,x_i) \partial_{ik} C(Y(t)).$$

*- Uniformly over any interval $[0,T]$, the kernel $\Upsilon^{(L)}$ has a deterministic and fixed limit $\lim_{n_{L-1} \to \infty} \cdots \lim_{n_1 \to \infty} \Upsilon_{kk'}^{(L)}(x,x') = \delta_{kk'} \Upsilon_\infty^{(L)}(x,x')$ with limiting kernel:*

$$\Upsilon_\infty^{(L)}(x,x') = \sum_{\ell=1}^{L-1} \left(\Theta_\infty^{(\ell)}(x,x')^2 \ddot{\Sigma}_\infty^{(\ell)}(x,x') + 2\Theta_\infty^{(\ell)}(x,x')\dot{\Sigma}_\infty^{(\ell)}(x,x')\right) \dot{\Sigma}_\infty^{(\ell+1)}(x,x') \cdots \dot{\Sigma}_\infty^{(L-1)}(x,x').$$

*- The higher moment $k > 2$ vanish:* $\lim_{n_{L-1} \to \infty} \cdots \lim_{n_1 \to \infty} \mathrm{Tr}\left(S^k\right) = 0$.

This result has a number of consequences for infinitely wide networks:

1. At initialization, the matrix $S$ has a finite Frobenius norm $\|S\|_F^2 = \mathrm{Tr}\left(S^2\right) = \nabla C^T \tilde{\Upsilon} \nabla C$, because $\Upsilon$ converges to a fixed limit. As the network converges, the derivative of the cost goes to zero $\nabla C(t) \to 0$ and so does the Frobenius norm of $S$.

2. In contrast the operator norm of $S$ vanishes already at initialization (because for all even $k$, we have $\|S\|_{op} \leq \sqrt[k]{\mathrm{Tr}\left(S^k\right)} \to 0$). At initialization, the vanishing of $S$ in operator norm but not in Frobenius norm can be explained by the matrix $S$ having a growing number of eigenvalues of shrinking intensity as the width grows.

3. When it comes to the first moment of $S$, Proposition 4 shows that the spectrum of $S$ is in general not symmetric. For the MSE loss the expectation of the first moment at initialization is

$$\mathbb{E}\left[\mathrm{Tr}(S)\right] = \mathbb{E}\left[(Y - Y^*)^T G\right] = \mathbb{E}\left[Y^T G\right] - (Y^*)^T \mathbb{E}\left[G\right] = \mathrm{Tr}\left(\tilde{\Phi}\right) - 0$$

which may be positive or negative depending on the choice of nonlinearity: with a smooth ReLU, it is positive, while for the arc-tangent or the normalized smooth ReLU, it can be negative (see Figure 1).

This is in contrast to the result obtained in (Pennington & Bahri, 2017; Geiger et al., 2018) for the shallow ReLU networks, taking the second derivative of the ReLU to be zero. Under this assumption the spectrum of $S$ is symmetric: if the eigenvalues are ordered from lowest to highest, $\lambda_i = -\lambda_{P-i}$ and $\mathrm{Tr}(S) = 0$.

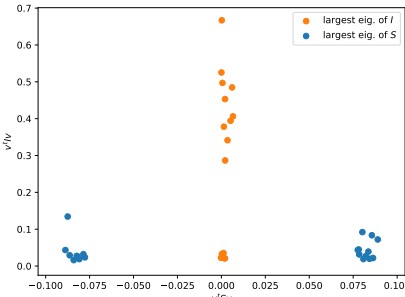

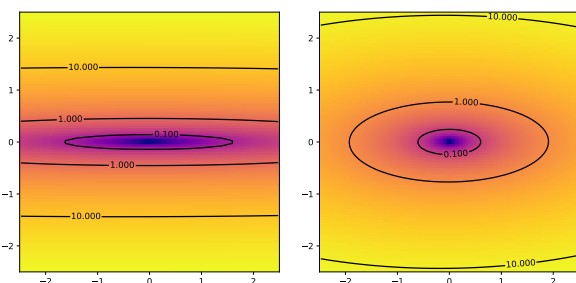

Figure 2: Illustration of the mutual orthogonality of $I$ and $S$. For the 20 first eigenvectors of $I$ (blue) and $S$ (orange), we plot the Rayleigh quotients $v^T I v$ and $v^T S v$ (with $L = 3$, $n_1 = n_2 = 1000$ and the normalized ReLU on 14x14 MNIST with $N = 256$). We see that the directions where $I$ is large are directions where $S$ is small and vice versa.

Figure 3: Plot of the loss surface around a global minimum along the first (along the y coordinate) and fourth (x coordinate) eigenvectors of $I$. The network has $L = 4$, width $n_1 = n_2 = n_3 = 1000$ for the smooth ReLU (left) and the normalized smooth ReLU (right). The data is uniform on the unit disk. Normalizing the non-linearity greatly reduces the narrow valley structure of the loss thus speeding up training.

These observations suggest that $S$ has little influence on the shape of the surface, especially towards the end of training, the matrix $I$ however has an interesting structure.

### 3.5 The matrix $I$

At a global minimizer $\theta^*$, the spectrum of $I$ describes how the loss behaves around $\theta^*$. Along the eigenvectors of the biggest eigenvalues of $I$, the loss increases rapidly, while small eigenvalues correspond to flat directions. Numerically, it has been observed that the matrix $I$ features a few dominating eigenvalues and a bulk of small eigenvalues (Sagun et al., 2016; 2017; Gur-Ari et al., 2018; Papyan, 2019). This leads to a narrow valley structure of the loss around a minimum: the biggest eigenvalues are the 'cliffs' of the valley, i.e. the directions along which the loss grows fastest, while the small eigenvalues form the 'flat directions' or the bottom of the valley.

Note that the rank of $I$ is bounded by $Nn_L$ and in the overparametrized regime, when $Nn_L < P$, the matrix $I$ will have a large nullspace, these are directions along which the value of the function on the training set does not change. Note that in the overparametrized regime, global minima are not isolated: they lie in a manifold of dimension at least $P - Nn_L$ and the nullspace of $I$ is tangent to this solution manifold.

The matrix $I$ is closely related to the NTK Gram matrix:

$$\tilde{\Theta} = \mathcal{D}Y^{(L)} \left( \mathcal{D}Y^{(L)} \right)^T \text{ and } I = \left( \mathcal{D}Y^{(L)} \right)^T \mathcal{H}C\mathcal{D}Y^{(L)}.$$

As a result, the limiting spectrum of the matrix $I$ can be directly obtained from the NTK[2]

**Proposition 1.** *For any loss $C$ with BGOSS and $\sigma \in C_b^4(\mathbb{R})$, uniformly over any interval $[0, T]$, the moments $\mathrm{Tr}\left(I^k\right)$ converge to the following limit (with the convention that $i_{k+1} = i_1$):*

$$\lim_{n_{L-1} \to \infty} \cdots \lim_{n_1 \to \infty} \mathrm{Tr}\left(I^k\right) = \mathrm{Tr}\left(\left(\mathcal{H}C(Y_t)\tilde{\Theta}\right)^k\right) = \frac{1}{N^k} \sum_{i_1,\ldots,i_k=1}^{N} \prod_{m=1}^{k} c_{i_m}''(f_{\theta(t)}(x_{i_m}))\Theta_\infty^{(L)}(x_{i_m}, x_{i_{m+1}})$$

---

[2]This result was already obtained in (Karakida et al., 2018), but without identifying the NTK explicitly and only at initialization.

*Proof.* It follows from $\text{Tr}\left(I^k\right) = \text{Tr}\left(\left(\left(\mathcal{D}Y^{(L)}\right)^T \mathcal{H}C\mathcal{D}Y^{(L)}\right)^k\right) = \text{Tr}\left(\left(\mathcal{H}C\tilde{\Theta}\right)^k\right)$ and the asymptotic of the NTK (Jacot et al., 2018). ☐

### 3.5.1 MEAN-SQUARE ERROR

When the loss is the MSE, $\mathcal{H}C$ is equal to $\frac{1}{N}Id_{Nn_L}$. As a result, $\tilde{\Theta}$ and $I$ have the same non-zero eigenvalues up to a scaling of $1/N$. Because the NTK is assymptotically fixed, the spectrum of $I$ is also fixed in the limit.

The eigenvectors of the NTK Gram matrix are the kernel principal components of the data. The biggest principal components are the directions in function space which are most favorised by the NTK. This gives a functional interpretation of the narrow valley structure in DNNs: the cliffs of the valley are the biggest principal components, while the flat directions are the smallest components.

*Remark* 2. As the depth $L$ of the network increases, one can observe two regimes (Poole et al., 2016; Jacot et al., 2019): Order/Freeze where the NTK converges to a constant and Chaos where the NTK converges to a Kronecker delta. In the Order/Freeze the $Nn_L \times Nn_L$ Gram matrix approaches a block diagonal matrix with $n_L$ constant blocks, and as a result $n_L$ eigenvalues of $I$ dominate the other ones, corresponding to constant directions along each outputs (this is in line with the observations of (Papyan, 2019)). This leads to a narrow valley for the loss and slows down training. In contrast, in the Chaos regime, the NTK Gram matrix approaches a scaled identity matrix, and the spectrum of $I$ should hence concentrate around a positive value, hence speeding up training. Figure 3 illustrates this phenomenon: with the smooth ReLU we observe a narrow valley, while with the normalized smooth ReLU (which lies in the Chaos according to (Jacot et al., 2019)) the narrowness of the loss is reduced. A similar phenomenon may explain why normalization helps smoothing the loss surface and speed up training (Santurkar et al., 2018; Ghorbani et al., 2019).

### 3.5.2 CROSS-ENTROPY LOSS

For a binary cross-entropy loss with labels $Y^* \in \{-1, +1\}^N$

$$C(Y) = \frac{1}{N}\sum_{i=1}^{N} log\left(1 + e^{-Y_i^* Y_i}\right),$$

$\mathcal{H}C$ is a diagonal matrix whose entries depend on $Y$ (but not on $Y^*$):

$$\mathcal{H}_{ii}C(Y) = \frac{1}{N}\frac{1}{1 + e^{-Y_i} + e^{Y_i}}.$$

The eigenvectors of $I$ then correspond to the weighted kernel principal component of the data. The positive weights $\frac{1}{1+e^{-Y_i}+e^{Y_i}}$ approach $1/3$ as $Y_i$ goes to 0, i.e. when it is close to the decision boundary from one class to the other, and as $Y_i \to \pm\infty$ the weight go to zero. The weights evolve in time through $Y_i$, the spectrum of $I$ is therefore not asymptotically fixed as in the MSE case, but the functional interpretation of the spectrum in terms of the kernel principal components remains.

## 4 CONCLUSION

We have given an explicit formula for the limiting moments of the Hessian of DNNs throughout training. We have used the common decomposition of the Hessian in two terms $I$ and $S$ and have shown that the two terms are asymptotically mutually orthogonal, such that they can be studied separately.

The matrix $S$ vanishes in Frobenius norm as the network converges and has vanishing operator norm throughout training. The matrix $I$ is arguably the most important as it describes the narrow valley structure of the loss around a global minimum. The eigendecomposition of $I$ is related to the (weighted) kernel principal components of the data w.r.t. the NTK.

## Acknowledgements

Clément Hongler acknowledges support from the ERC SG CONSTAMIS grant, the NCCR SwissMAP grant, the NSF DMS-1106588 grant, the Minerva Foundation, the Blavatnik Family Foundation, and the Latsis foundation.

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

## A  PROOFS

For the proofs of the theorems and propositions presented in the main text, we reformulate
the setup of (Jacot et al., 2018). For a fixed training set $x_1, ..., x_N$, we consider a (possibly
random) time-varying training direction $D(t) \in \mathbb{R}^{N n_L}$ which describes how each of the
outputs must be modified. In the case of gradient descent on a cost $C(Y)$, the training
direction is $D(t) = \nabla C(Y(t))$. The parameters are updated according to the differential
equation

$$\partial_t \theta(t) = (\partial_\theta Y(t))^T D(t).$$

Under the condition that $\int_0^T \|D(t)\|_2 \, dt$ is stochastically bounded as the width of the network
goes to infinity, the NTK $\Theta^{(L)}$ converges to its fixed limit uniformly over $[0, T]$.

The reason we consider a general training direction (and not only a gradient of a loss) is
that we can split a network in two at a layer $\ell$ and the training of the smaller network will
be according to the training direction $D_i^{(\ell)}(t)$ given by

$$D_i^{(\ell)}(t) = diag\left(\dot{\sigma}\left(\alpha^{(\ell)}(x_i)\right)\right)\left(\frac{1}{\sqrt{n_\ell}}W^{(\ell)}\right)^T ...diag\left(\dot{\sigma}\left(\alpha^{(L-1)}(x_i)\right)\right)\left(\frac{1}{\sqrt{n_{L-1}}}W^{(L-1)}\right)^T D_i(t)$$

because the derivatives $\dot{\sigma}$ are bounded and by Lemma 1 of the Appendix of (Jacot et al.,
2018), this training direction satisfies the constraints even though it is not the gradient of a
loss. As a consequence, as $n_1 \to \infty, ..., n_{\ell-1} \to \infty$ the NTK of the smaller network $\Theta^{(\ell)}$ also
converges to its limit uniformly over $[0, T]$. As we let $n_\ell \to \infty$ the pre-activations $\tilde{\alpha}_i^{(\ell)}$ and
weights $W_{ij}^{(\ell)}$ move at a rate of $1/\sqrt{n_\ell}$. We will use this rate of change to prove that other
types of kernels are constant during training.

When a network is trained with gradient descent on a loss $C$ with BGOSS, the integral $\int_0^T \|D(t)\|_2 \, dt$ is stochastically bounded. Because the loss is decreasing during training, the outputs $Y(t)$ lie in the sublevel set $U_{C(Y(0))}$ for all times $t$. The norm of the gradient is hence bounded for all times $t$. Because the distribution of $Y(0)$ converges to a multivariate Gaussian, $b(C(Y(0)))$ is stochastically bounded as the width grows, where $b(a)$ is a bound on the norm of the gradient on $U_a$. We then have the bound $\int_0^T \|D(t)\|_2 \, dt \leq T b(C(Y(0)))$ which is itself stochastically bounded.

For the binary and softmax cross-entropy losses the gradient is uniformly bounded:

**Proposition 2.** *For the binary cross-entropy loss $C$ and any $Y \in \mathbb{R}^N$, $\|\nabla C(Y)\|_2 \leq \frac{1}{\sqrt{N}}$.*

*For the softmax cross-entropy loss $C$ on $c \in \mathbb{N}$ classes and any $Y \in \mathbb{R}^{Nc}$, $\|\nabla C(Y)\|_2 \leq \frac{\sqrt{2c}}{\sqrt{N}}$.*

*Proof.* The binary cross-entropy loss with labels $Y^* \in \{0, 1\}^N$ is

$$C(Y) = -\frac{1}{N} \sum_{i=1}^N \log \frac{e^{Y_i Y_i^*}}{1 + e^{Y_i}} = \frac{1}{N} \sum_{i=1}^N \log\left(1 + e^{Y_i}\right) - Y_i Y_i^*$$

and the gradient at an input $i$ is

$$\partial_i C(Y) = \frac{1}{N} \frac{e^{Y_i} - Y_i^*(1 + e^{Y_i})}{1 + e^{Y_i}}$$

which is bounded in absolute value by $\frac{1}{N}$ for both $Y_i^* = 0, 1$ such that $\|\nabla C(Y)\|_2 \leq \frac{1}{\sqrt{N}}$.

The softmax cross-entropy loss over $c$ classes with labels $Y^* \in \{1, \ldots, c\}^N$ is defined by

$$C(Y) = -\frac{1}{N} \sum_{i=1}^N \log \frac{e^{Y_i Y_i^*}}{\sum_{k=1}^c e^{Y_{ik}}} = \frac{1}{N} \sum_{i=1}^N \log\left(\sum_{k=1}^c e^{Y_{ik}}\right) - Y_{i Y_i^*}.$$

The gradient is at an input $i$ and output class $m$ is

$$\partial_{im} C(Y) = \frac{1}{N} \left(\frac{e^{Y_{im}}}{\sum_{k=1}^c e^{Y_{ik}}} - \delta_{Y_i^* m}\right)$$

which is bounded in absolute value by $\frac{2}{N}$ such that $\|\nabla C(Y)\|_2 \leq \frac{\sqrt{2c}}{\sqrt{N}}$. $\qquad\square$

## B  PRELIMINARIES

To study the moments of the matrix $S$, we first have to show that two tensors vanish as $n_1, ..., n_{L-1} \to \infty$:

$$\Omega_{k_0, k_1, k_2}^{(L)}(x_0, x_1, x_2) = (\nabla f_{\theta, k_0}(x_0))^T \mathcal{H} f_{\theta, k_1}(x_1) \nabla f_{\theta, k_2}(x_2)$$

$$\Gamma_{k_0, k_1, k_2, k_3}^{(L)}(x_0, x_1, x_2, x_4) = (\nabla f_{\theta, k_0}(x_0))^T \mathcal{H} f_{\theta, k_1}(x_1) \mathcal{H} f_{\theta, k_2}(x_2) \nabla f_{\theta, k_3}(x_3).$$

We study these tensors recursively, for this, we need a recursive definition for the first derivatives $\partial_{\theta_p} f_{\theta, k}(x)$ and second derivatives $\partial^2_{\theta_p \theta_{p'}} f_{\theta, k}(x)$. The value of these derivatives depend on the layer $\ell$ the parameters $\theta_p$ and $\theta_{p'}$ belong to, and on whether they are connection weights $W_{mk}^{(\ell)}$ or biases $b_k^{(\ell)}$. The derivatives with respect to the parameters of the last layer are

$$\partial_{W_{mk}^{(L-1)}} f_{\theta, k'}(x) = \frac{1}{\sqrt{n_{L-1}}} \alpha_m^{(L-1)}(x) \delta_{kk'}$$

$$\partial_{b_k^{(L-1)}} f_{\theta, k'}(x) = \beta^2 \delta_{kk'}$$

for parameters $\theta_p$ which belong to the lower layers the derivatives can be defined recursively by

$$\partial_{\theta_p} f_{\theta, k}(x) = \frac{1}{\sqrt{n_{L-1}}} \sum_{m=1}^{n_{L-1}} \partial_{\theta_p} \tilde{\alpha}_m^{(L-1)}(x) \dot{\sigma}\left(\tilde{\alpha}_m^{(L-1)}(x)\right) W_{mk}^{(L-1)}.$$

For the second derivatives, we first note that if either of the parameters $\theta_p$ or $\theta_{p'}$ are bias of the last layer, or if they are both connection weights of the last layer, then $\partial^2_{\theta_p \theta_{p'}} f_{\theta,k}(x) = 0$. Two cases are left: when one parameter is a connection weight of the last layer and the others belong to the lower layers, and when both belong to the lower layers. Both cases can be defined recursively in terms of the first and second derivatives of $\tilde{\alpha}_m^{(L-1)}$:

$$\partial^2_{\theta_p W_{mk}^{(L)}} f_{\theta,k'}(x) = \frac{1}{\sqrt{n_{L-1}}} \partial_{\theta_p} \tilde{\alpha}_m^{(L-1)}(x) \dot{\sigma}\left(\tilde{\alpha}_m^{(L-1)}(x)\right) \delta_{kk'}$$

$$\partial^2_{\theta_p \theta_{p'}} f_{\theta,k'}(x) = \frac{1}{\sqrt{n_{L-1}}} \sum_{m=1}^{n_{L-1}} \partial^2_{\theta_p \theta_{p'}} \tilde{\alpha}_m^{(L-1)}(x) \dot{\sigma}\left(\tilde{\alpha}_m^{(L-1)}(x)\right) W_{mk}^{(L-1)}$$

$$+ \frac{1}{\sqrt{n_{L-1}}} \sum_{m=1}^{n_{L-1}} \partial_{\theta_p} \tilde{\alpha}_m^{(L-1)}(x) \partial_{\theta_{p'}} \tilde{\alpha}_m^{(L-1)}(x) \ddot{\sigma}\left(\tilde{\alpha}_m^{(L-1)}(x)\right) W_{mk}^{(L-1)}.$$

Using these recursive definitions, the tensors $\Omega^{(L+1)}$ and $\Gamma^{(L+1)}$ are given in terms of $\Theta^{(L)}, \Omega^{(L)}$ and $\Gamma^{(L)}$, in the same manner that the NTK $\Theta^{(L+1)}$ is defined recursively in terms of $\Theta^{(L)}$ in (Jacot et al., 2018).

**Lemma 1.** *For any loss $C$ with BGOSS and $\sigma \in C_b^4(\mathbb{R})$, we have uniformly over $[0, T]$*

$$\lim_{n_{L-1} \to \infty} \cdots \lim_{n_1 \to \infty} \Omega_{k_0,k_1,k_2}^{(L)}(x_0, x_1, x_2) = 0$$

*Proof.* The proof is done by induction. When $L = 1$ the second derivatives $\partial^2_{\theta_p \theta_{p'}} f_{\theta,k}(x) = 0$ and $\Omega_{k_0,k_1,k_2}^{(L)}(x_0, x_1, x_2) = 0$.

For the induction step, we write $\Omega_{k_0,k_1,k_2}^{(\ell+1)}(x_0, x_1, x_2)$ recursively as

$$n_\ell^{-3/2} \sum_{m_0,m_1,m_2} \Theta_{m_0,m_1}^{(\ell)}(x_0, x_1) \Theta_{m_1,m_2}^{(\ell)}(x_1, x_2) \dot{\sigma}(\tilde{\alpha}_{m_0}^{(\ell)}(x_0)) \ddot{\sigma}(\tilde{\alpha}_{m_1}^{(\ell)}(x_1)) \dot{\sigma}(\tilde{\alpha}_{m_2}^{(\ell)}(x_2)) W_{m_0 k_0}^{(\ell)} W_{m_1 k_1}^{(\ell)} W_{m_2 k_2}^{(\ell)}$$

$$+ n_\ell^{-3/2} \sum_{m_0,m_1,m_2} \Omega_{m_0,m_1,m_2}^{(\ell)}(x_0, x_1, x_2) \dot{\sigma}(\tilde{\alpha}_{m_0}^{(\ell)}(x_0)) \dot{\sigma}(\tilde{\alpha}_{m_1}^{(\ell)}(x_1)) \dot{\sigma}(\tilde{\alpha}_{m_2}^{(\ell)}(x_2)) W_{m_0 k_0}^{(\ell)} W_{m_1 k_1}^{(\ell)} W_{m_2 k_2}^{(\ell)}$$

$$+ n_\ell^{-3/2} \sum_{m_0,m_1} \Theta_{m_0,m_1}^{(\ell)}(x_0, x_1) \dot{\sigma}(\tilde{\alpha}_{m_0}^{(\ell)}(x_0)) \dot{\sigma}(\tilde{\alpha}_{m_1}^{(\ell)}(x_1)) \sigma(\tilde{\alpha}_{m_1}^{(\ell)}(x_2)) W_{m_0 k_0}^{(\ell)} \delta_{k_1 k_2}$$

$$+ n_\ell^{-3/2} \sum_{m_1,m_2} \Theta_{m_1,m_2}^{(\ell)}(x_1, x_2) \sigma(\tilde{\alpha}_{m_1}^{(\ell)}(x_0)) \dot{\sigma}(\tilde{\alpha}_{m_1}^{(\ell)}(x_1)) \dot{\sigma}(\tilde{\alpha}_{m_2}^{(\ell)}(x_2)) \delta_{k_0 k_1} W_{m_2 k_2}^{(\ell)}.$$

As $n_1, ..., n_{\ell-1} \to \infty$ and for any times $t < T$, the NTK $\Theta^{(\ell)}$ converges to its limit while $\Omega^{(\ell)}$ vanishes. The second summand hence vanishes and the others converge to

$$n_\ell^{-3/2} \sum_m \Theta_\infty^{(\ell)}(x_0, x_1) \Theta_\infty^{(\ell)}(x_1, x_2) \dot{\sigma}(\tilde{\alpha}_m^{(\ell)}(x_0)) \ddot{\sigma}(\tilde{\alpha}_m^{(\ell)}(x_1)) \dot{\sigma}(\tilde{\alpha}_m^{(\ell)}(x_2)) W_{m k_0}^{(\ell)} W_{m k_1}^{(\ell)} W_{m k_2}^{(\ell)}$$

$$+ n_\ell^{-3/2} \sum_m \Theta_\infty^{(\ell)}(x_0, x_1) \dot{\sigma}(\tilde{\alpha}_m^{(\ell)}(x_0)) \dot{\sigma}(\tilde{\alpha}_m^{(\ell)}(x_1)) \sigma(\tilde{\alpha}_m^{(\ell)}(x_2)) W_{m k_0}^{(\ell)} \delta_{k_1 k_2}$$

$$+ n_\ell^{-3/2} \sum_m \Theta_\infty^{(\ell)}(x_1, x_2) \sigma(\tilde{\alpha}_m^{(\ell)}(x_0)) \dot{\sigma}(\tilde{\alpha}_m^{(\ell)}(x_1)) \dot{\sigma}(\tilde{\alpha}_m^{(\ell)}(x_2)) \delta_{k_0 k_1} W_{m k_2}^{(\ell)}.$$

At initialization, all terms vanish as $n_\ell \to \infty$ because all summands are independent with zero mean and finite variance: in the $n_1 \to \infty, \ldots, n_{\ell-1} \to \infty$ limit, the $\tilde{\alpha}_m^{(\ell)}(x)$ are independent for different $m$, see (Jacot et al., 2018). During training, the weights $W^{(\ell)}$ and preactivations $\tilde{\alpha}^{(\ell)}$ move at a rate of $1/\sqrt{n_\ell}$ (see the proof of convergence of the NTK in (Jacot et al., 2018)). Since $\dot{\sigma}$ is Lipschitz, we obtain that the motion during training of each of the sums is of order $n_\ell^{-3/2+1/2} = n_\ell^{-1}$. As a result, uniformly over times $t \in [0, T]$, all the sums vanish. $\square$

Similarily, we have

**Lemma 2.** *For any loss $C$ with BGOSS and $\sigma \in C_b^4(\mathbb{R})$, we have uniformly over $[0, T]$*

$$\lim_{n_{L-1} \to \infty} \cdots \lim_{n_1 \to \infty} \Gamma_{k_0, k_1, k_2, k_3}^{(L)}(x_0, x_1, x_2, x_3) = 0$$

*Proof.* The proof is done by induction. When $L = 1$ the hessian $\mathcal{H}F^{(1)} = 0$, such that $\Gamma_{k_0, k_1, k_2, k_3}^{(L)}(x_0, x_1, x_2, x_3) = 0$.

For the induction step, $\Gamma^{(\ell+1)}$ can be defined recursively:

$$\Gamma_{k_0, k_1, k_2, k_3}^{(L+1)}(x_0, x_1, x_2, x_3)$$

$$= n_L^{-2} \sum_{m_0, m_1, m_2, m_3} \Gamma_{m_0, m_1, m_2, m_3}^{(L)}(x_0, x_1, x_2, x_3) \dot{\sigma}(\alpha_{m_0}^{(L)}(x_0)) \dot{\sigma}(\alpha_{m_1}^{(L)}(x_1)) \dot{\sigma}(\alpha_{m_2}^{(L)}(x_2)) \dot{\sigma}(\alpha_{m_3}^{(L)}(x_3))$$

$$W_{m_0 k_0}^{(L)} W_{m_1 k_1}^{(L)} W_{m_2 k_2}^{(L)} W_{m_3 k_3}^{(L)}$$

$$+ n_L^{-2} \sum_{m_0, m_1, m_2, m_3} \Theta_{m_0, m_1}^{(L)}(x_0, x_1) \Omega_{m_1, m_2, m_3}^{(L)}(x_1, x_2, x_3) \dot{\sigma}(\alpha_{m_0}^{(L)}(x_0)) \ddot{\sigma}(\alpha_{m_1}^{(L)}(x_1))$$

$$\dot{\sigma}(\alpha_{m_2}^{(L)}(x_2)) \dot{\sigma}(\alpha_{m_3}^{(L)}(x_3)) W_{m_0 k_0}^{(L)} W_{m_1 k_1}^{(L)} W_{m_2 k_2}^{(L)} W_{m_3 k_3}^{(L)}$$

$$+ n_L^{-2} \sum_{m_0, m_1, m_2, m_3} \Omega_{m_0, m_1, m_2}^{(L)}(x_0, x_1, x_2) \Theta_{m_2, m_3}^{(L)}(x_2, x_3) \dot{\sigma}(\alpha_{m_0}^{(L)}(x_0)) \dot{\sigma}(\alpha_{m_1}^{(L)}(x_1))$$

$$\ddot{\sigma}(\alpha_{m_2}^{(L)}(x_2)) \dot{\sigma}(\alpha_{m_3}^{(L)}(x_3)) W_{m_0 k_0}^{(L)} W_{m_1 k_1}^{(L)} W_{m_2 k_2}^{(L)} W_{m_3 k_3}^{(L)}$$

$$+ n_L^{-2} \sum_{m_0, m_1, m_2, m_3} \Theta_{m_0, m_1}^{(L)}(x_0, x_1) \Theta_{m_1, m_2}^{(L)}(x_1, x_2) \Theta_{m_2, m_3}^{(L)}(x_2, x_3) \dot{\sigma}(\alpha_{m_0}^{(L)}(x_0)) \ddot{\sigma}(\alpha_{m_1}^{(L)}(x_1))$$

$$\ddot{\sigma}(\alpha_{m_2}^{(L)}(x_2)) \dot{\sigma}(\alpha_{m_3}^{(L)}(x_3)) W_{m_0 k_0}^{(L)} W_{m_1 k_1}^{(L)} W_{m_2 k_2}^{(L)} W_{m_3 k_3}^{(L)}$$

$$+ n_L^{-2} \sum_{m_1, m_2, m_3} \Omega_{m_1, m_2, m_3}^{(L)}(x_1, x_2, x_3) \sigma(\alpha_{m_1}^{(L)}(x_0)) \dot{\sigma}(\alpha_{m_1}^{(L)}(x_1)) \dot{\sigma}(\alpha_{m_2}^{(L)}(x_2)) \dot{\sigma}(\alpha_{m_3}^{(L)}(x_3))$$

$$\delta_{k_0 k_1} W_{m_2 k_2}^{(L)} W_{m_3 k_3}^{(L)}$$

$$+ n_L^{-2} \sum_{m_1, m_2, m_3} \Theta_{m_1, m_2}^{(L)}(x_1, x_2) \Theta_{m_2, m_3}^{(L)}(x_2, x_3) \sigma(\alpha_{m_1}^{(L)}(x_0)) \dot{\sigma}(\alpha_{m_1}^{(L)}(x_1)) \ddot{\sigma}(\alpha_{m_2}^{(L)}(x_2)) \dot{\sigma}(\alpha_{m_3}^{(L)}(x_3))$$

$$\delta_{k_0 k_1} W_{m_2 k_2}^{(L)} W_{m_3 k_3}^{(L)}$$

$$+ n_L^{-2} \sum_{m_0, m_1, m_2} \Omega_{m_0, m_1, m_2}^{(L)}(x_0, x_1, x_2) \dot{\sigma}(\alpha_{m_0}^{(L)}(x_0)) \dot{\sigma}(\alpha_{m_1}^{(L)}(x_1)) \dot{\sigma}(\alpha_{m_2}^{(L)}(x_2)) \sigma(\alpha_{m_2}^{(L)}(x_3))$$

$$W_{m_0 k_0}^{(L)} W_{m_1 k_1}^{(L)} \delta_{k_2 k_3}$$

$$+ n_L^{-2} \sum_{m_0, m_1, m_2} \Theta_{m_0, m_1}^{(L)}(x_0, x_1) \Theta_{m_1, m_2}^{(L)}(x_1, x_2) \dot{\sigma}(\alpha_{m_0}^{(L)}(x_0)) \ddot{\sigma}(\alpha_{m_1}^{(L)}(x_1)) \dot{\sigma}(\alpha_{m_2}^{(L)}(x_2)) \sigma(\alpha_{m_2}^{(L)}(x_3))$$

$$W_{m_0 k_0}^{(L)} W_{m_1 k_1}^{(L)} \delta_{k_2 k_3}$$

$$+ n_L^{-2} \sum_{m_1, m_2} \Theta_{m_1, m_2}^{(L)}(x_1, x_2) \sigma(\alpha_{m_1}^{(L)}(x_0)) \dot{\sigma}(\alpha_{m_1}^{(L)}(x_1)) \dot{\sigma}(\alpha_{m_2}^{(L)}(x_2)) \sigma(\alpha_{m_2}^{(L)}(x_3)) \delta_{k_0 k_1} \delta_{k_2 k_3}$$

$$+ n_L^{-2} \sum_{m_0, m_1, m_3} \Theta_{m_0, m_1}^{(L)}(x_0, x_1) \Theta_{m_1, m_3}^{(L)}(x_2, x_3) \dot{\sigma}(\alpha_{m_0}^{(L)}(x_0)) \dot{\sigma}(\alpha_{m_1}^{(L)}(x_1)) \dot{\sigma}(\alpha_{m_1}^{(L)}(x_2)) \dot{\sigma}(\alpha_{m_3}^{(L)}(x_3))$$

$$W_{m_0 k_0}^{(L)} \delta_{k_1 k_2} W_{m_3 k_3}^{(L)}$$

As $n_1, ..., n_{\ell-1} \to \infty$ and for any times $t < T$, the NTK $\Theta^{(\ell)}$ converges to its limit while $\Omega^{(\ell)}$ and $\Gamma^{(\ell)}$ vanishes. $\Gamma_{k_0, k_1, k_2, k_3}^{(L+1)}(x_0, x_1, x_2, x_3)$ therefore converges to:

$$+ n_L^{-2} \sum_m \Theta_\infty^{(L)}(x_0, x_1) \Theta_\infty^{(L)}(x_1, x_2) \Theta_\infty^{(L)}(x_2, x_3) \dot{\sigma}(\alpha_m^{(L)}(x_0)) \ddot{\sigma}(\alpha_m^{(L)}(x_1)) \ddot{\sigma}(\alpha_m^{(L)}(x_2)) \dot{\sigma}(\alpha_m^{(L)}(x_3))$$

$$W_{m k_0}^{(L)} W_{m k_1}^{(L)} W_{m k_2}^{(L)} W_{m k_3}^{(L)}$$

$$+n_L^{-2} \sum_m \Theta_\infty^{(L)}(x_1, x_2)\Theta_\infty^{(L)}(x_2, x_3)\sigma(\alpha_m^{(L)}(x_0))\dot\sigma(\alpha_m^{(L)}(x_1))\ddot\sigma(\alpha_m^{(L)}(x_2))\dot\sigma(\alpha_m^{(L)}(x_3))$$

$$\delta_{k_0 k_1} W_{mk_2}^{(L)} W_{mk_3}^{(L)}$$

$$+n_L^{-2} \sum_m \Theta_\infty^{(L)}(x_0, x_1)\Theta_\infty^{(L)}(x_1, x_2)\dot\sigma(\alpha_m^{(L)}(x_0))\ddot\sigma(\alpha_m^{(L)}(x_1))\dot\sigma(\alpha_m^{(L)}(x_2))\sigma(\alpha_m^{(L)}(x_3))$$

$$W_{mk_0}^{(L)} W_{mk_1}^{(L)} \delta_{k_2 k_3}$$

$$+n_L^{-2} \sum_m \Theta_\infty^{(L)}(x_1, x_2)\sigma(\alpha_m^{(L)}(x_0))\dot\sigma(\alpha_m^{(L)}(x_1))\dot\sigma(\alpha_m^{(L)}(x_2))\sigma(\alpha_m^{(L)}(x_3))\delta_{k_0 k_1}\delta_{k_2 k_3}$$

$$+n_L^{-2} \sum_m \Theta_\infty^{(L)}(x_0, x_1)\Theta_\infty^{(L)}(x_2, x_3)\dot\sigma(\alpha_m^{(L)}(x_0))\dot\sigma(\alpha_m^{(L)}(x_1))\dot\sigma(\alpha_m^{(L)}(x_2))\dot\sigma(\alpha_m^{(L)}(x_3))$$

$$W_{mk_0}^{(L)} \delta_{k_1 k_2} W_{mk_3}^{(L)}$$

For the convergence during training, we proceed similarily to the proof of Lemma 1. At initialization, all terms vanish as $n_\ell \to \infty$ because all summands are independent (after taking the $n_1, \ldots, n_{L-1} \to \infty$ limit) with zero mean and finite variance. During training, the weights $W^{(\ell)}$ and preactivations $\tilde\alpha^{(\ell)}$ move at a rate of $1/\sqrt{n_\ell}$ which leads to a change of order $n_\ell^{-2+1/2} = n_\ell^{-1.5}$, which vanishes for all times $t$ too. $\qquad\square$

## C    THE MATRIX $S$

We now have the theoretical tools to describe the moments of the matrix $S$. We first give a bound for the rank of $S$:

**Proposition 3.** $Rank(S) \leq 2(n_1 + \ldots + n_{L-1})Nn_L$

*Proof.* We first observe that $S$ is given by a sum of $Nn_L$ matrices:

$$S_{pp'} = \sum_{i=1}^{N} \sum_{k=1}^{n_L} \partial_{ik} C \partial_{\theta_p \theta_p}^2 f_{\theta,k}(x_i).$$

It is therefore sufficiant to show that the rank of each matrices $\mathcal{H}f_{\theta,k}(x) = \left(\partial_{\theta_p \theta_{p'}}^2 f_{\theta,k}(x_i)\right)_{p,p'}$ is bounded by $2(n_1 + \ldots + n_L)$.

The derivatives $\partial_{\theta_p} f_{\theta,k}(x)$ have different definition depending on whether the parameter $\theta_p$ is a connection weight $W_{ij}^{(\ell)}$ or a bias $b_j^{(\ell)}$:

$$\partial_{W_{ij}^{(\ell)}} f_{\theta,k}(x) = \frac{1}{\sqrt{n_\ell}} \alpha_i^{(\ell)}(x;\theta)\partial_{\tilde\alpha_j^{(\ell+1)}(x;\theta)} f_{\theta,k}(x)$$

$$\partial_{b_j^{(\ell)}} f_{\theta,k}(x) = \beta\partial_{\tilde\alpha_j^{(\ell+1)}(x;\theta)} f_{\theta,k}(x)$$

These formulas only depend on $\theta$ through the values $\left(\alpha_i^{(\ell)}(x;\theta)\right)_{\ell,i}$ and $\left(\partial_{\tilde\alpha_i^{(\ell)}(x;\theta)} f_{\theta,k}(x)\right)_{\ell,i}$ for $\ell = 1, \ldots, L-1$ (note that both $\alpha_i^{(0)}(x) = x_i$ and $\partial_{\tilde\alpha_i^{(L)}(x;\theta)} f_{\theta,k}(x) = \delta_{ik}$ do not depend on $\theta$). Together there are $2(n_1 + \ldots + n_{L-1})$ of them. As a consequence, the map $\theta \mapsto \left(\partial_{\theta_p} f_{\theta,k}(x_i)\right)_p$ can be written as a composition

$$\theta \in \mathbb{R}^P \mapsto \left(\alpha_i^{(\ell)}(x;\theta), \partial_{\tilde\alpha_i^{(\ell)}(x;\theta)} f_{\theta,k}(x)\right)_{\ell,i} \in \mathbb{R}^{2(n_1+\ldots+n_{L-1})} \mapsto \left(\partial_{\theta_p} f_{\theta,k}(x_i)\right)_p \in \mathbb{R}^P$$

and the matrix $\mathcal{H}f_{\theta,k}(x)$ is equal to the Jacobian of this map. By the chain rule, $\mathcal{H}f_{\theta,k}(x)$ is the matrix multiplication of the Jacobians of the two submaps, whose rank are bounded by $2(n_1 + \ldots + n_{L-1})$, hence bounding the rank of $\mathcal{H}f_{\theta,k}(x)$. And because $S$ is a sum of $Nn_L$ matrices of rank smaller than $2(n_1 + \ldots + n_{L-1})$, the rank of $S$ is bounded by $2(n_1 + \ldots + n_{L-1})Nn_L$. $\qquad\square$

### C.1 Moments

Let us now prove Proposition 4:

**Proposition 4.** *For any loss $C$ with BGOSS and $\sigma \in C_b^4(\mathbb{R})$, the first two moments of $S$ take the form*

$$\mathrm{Tr}\,(S(t)) = G(t)^T \nabla C(t)$$
$$\mathrm{Tr}\,(S(t)^2) = \nabla C(t)^T \tilde{\Upsilon}(t) \nabla C(t)$$

*- At initialization, $g_\theta$ and $f_\theta$ converge to a (centered) Gaussian pair with covariances*

$$\mathbb{E}[g_{\theta,k}(x)g_{\theta,k'}(x')] = \delta_{kk'}\Xi_\infty^{(L)}(x,x')$$
$$\mathbb{E}[g_{\theta,k}(x)f_{\theta,k'}(x')] = \delta_{kk'}\Phi_\infty^{(L)}(x,x')$$
$$\mathbb{E}[f_{\theta,k}(x)f_{\theta,k'}(x')] = \delta_{kk'}\Sigma_\infty^{(L)}(x,x')$$

*and during training $g_\theta$ evolves according to*

$$\partial_t g_{\theta,k}(x) = \sum_{i=1}^N \Lambda_\infty^{(L)}(x,x_i)\partial_{ik}C(Y(t)).$$

*- Uniformly over any interval $[0,T]$ where $\int_0^T \|\nabla C(t)\|_2\, dt$ is stochastically bounded, the kernel $\Upsilon^{(L)}$ has a deterministic and fixed limit $\lim_{n_{L-1}\to\infty}\cdots\lim_{n_1\to\infty}\Upsilon_{kk'}^{(L)}(x,x') = \delta_{kk'}\Upsilon_\infty^{(L)}(x,x')$ with limiting kernel:*

$$\Upsilon_\infty^{(L)}(x,x') = \sum_{\ell=1}^{L-1}\left(\Theta_\infty^{(\ell)}(x,x')^2\ddot{\Sigma}^{(\ell)}(x,x') + 2\Theta_\infty^{(\ell)}(x,x')\dot{\Sigma}^{(\ell)}(x,x')\right)\dot{\Sigma}^{(\ell+1)}(x,x')\cdots\dot{\Sigma}^{(L-1)}(x,x').$$

*- The higher moment $k > 2$ vanish: $\lim_{n_{L-1}\to\infty}\cdots\lim_{n_1\to\infty}\mathrm{Tr}\,(S^k) = 0$.*

*Proof.* The first moment of $S$ takes the form

$$\mathrm{Tr}\,(S) = \sum_p (\nabla C)^T \mathcal{H}_{p,p} Y = (\nabla C)^T G$$

where $G$ is the restriction to the training set of the function $g_\theta(x) = \sum_p \partial_{\theta_p\theta_p}^2 f_\theta(x)$. This process is random at initialization and varies during training. Lemma 3 below shows that, in the infinite width limit, it is a Gaussian process at initialization which then evolves according to a simple differential equation, hence describing the evolution of the first moment during training.

The second moment of $S$ takes the form:

$$\mathrm{Tr}(S^2) = \sum_{p_1,p_2=1}^P \sum_{i_1,i_2=1}^N \partial_{\theta_{p_1},\theta_{p_2}}^2 f_{\theta,k_1}(x_1)\partial_{\theta_{p_2},\theta_{p_1}}^2 f_{\theta,k_2}(x_2)c_{i_1}'(x_{i_1})c_{i_2}'(x_{i_2})$$
$$= (\nabla C)^T \tilde{\Upsilon} \nabla C$$

where $\Upsilon_{k_1,k_2}^{(L)}(x_1,x_2) = \sum_{p_1,p_2=1}^P \partial_{\theta_{p_1},\theta_{p_2}}^2 f_{\theta,k_1}(x_1)\partial_{\theta_{p_2},\theta_{p_1}}^2 f_{\theta,k_2}(x_2)$ is a multidimensional kernel and $\tilde{\Upsilon}$ is its Gram matrix. Lemma 4 below shows that in the infinite-width limit, $\Upsilon_{k_1,k_2}^{(L)}(x_1,x_2)$ converges to a deterministic and time-independent limit $\Upsilon_\infty^{(L)}(x_1,x_2)\delta_{k_1k_2}$.

To show that $\mathrm{Tr}(S^k) \to 0$ for all $k > 2$, it suffices to show that $\|S^2\|_F \to 0$ as $|\mathrm{Tr}(S^k)| < \|S^2\|_F \|S\|_F^{k-2}$ and we know that $\|S\|_F \to (\partial_Y C)^T \tilde{\Upsilon}\partial_Y C$ is finite. We have that

$$\|S^2\|_F = \sum_{i_0,i_1,i_2,i_3=1}^N \sum_{k_0,k_1,k_2,k_3=1}^{n_L} \Psi_{k_0,k_1,k_2,k_3}^{(L)}(x_{i_0},x_{i_1},x_{i_2},x_{i_3})\partial_{f_{\theta,k_0}(x_{i_0})}C\partial_{f_{\theta,k_1}(x_{i_1})}C$$

$$\partial_{f_{\theta,k_2}(x_{i_2})}C\partial_{f_{\theta,k_3}(x_{i_3})}C$$

$$= \tilde{\Psi} \cdot (\partial_Y C)^{\otimes 4}$$

for $\tilde{\Psi}$ the $Nn_L \times Nn_L \times Nn_L \times Nn_L$ finite version of

$$\Psi^{(L)}_{k_0,k_1,k_2,k_3}(x_{i_0}, x_{i_1}, x_{i_2}, x_{i_3}) = \sum_{p_0,p_1,p_2,p_3=1}^{P} \partial^2_{\theta_{p_0},\theta_{p_1}} f_{\theta,k_0}(x_0)\partial^2_{\theta_{p_1},\theta_{p_2}} f_{\theta,k_1}(x_1)$$

$$\partial^2_{\theta_{p_2},\theta_{p_3}} f_{\theta,k_2}(x_2)\partial^2_{\theta_{p_3},\theta_{p_0}} f_{\theta,k_3}(x_3).$$

which vanishes in the infinite width limit by Lemma 5 below. □

**Lemma 3.** *For any loss $C$ with BGOSS and $\sigma \in C_b^4(\mathbb{R})$, at initialization $g_\theta$ and $f_\theta$ converge to a (centered) Gaussian pair with covariances*

$$\mathbb{E}[g_{\theta,k}(x)g_{\theta,k'}(x')] = \delta_{kk'}\Xi^{(L)}_\infty(x,x')$$
$$\mathbb{E}[g_{\theta,k}(x)f_{\theta,k'}(x')] = \delta_{kk'}\Phi^{(L)}_\infty(x,x')$$
$$\mathbb{E}[f_{\theta,k}(x)f_{\theta,k'}(x')] = \delta_{kk'}\Sigma^{(L)}_\infty(x,x')$$

*and during training $g_\theta$ evolves according to*

$$\partial_t g_\theta(x) = \sum_{i=1}^{N} \Lambda^{(L)}_\infty(x,x_i)D_i(t)$$

*Proof.* When $L = 1$, $g_\theta(x)$ is 0 for any $x$ and $\theta$.

For the inductive step, the trace $g^{(L+1)}_{\theta,k}(x)$ is defined recursively as

$$\frac{1}{\sqrt{n_L}} \sum_{m=1}^{n_L} g^{(L)}_{\theta,m}(x)\dot{\sigma}\left(\tilde{\alpha}^{(L)}_m(x)\right) W^{(L)}_{mk} + \mathrm{Tr}\left(\nabla f_{\theta,m}(x)\left(\nabla f_{\theta,m}(x)\right)^T\right)\ddot{\sigma}\left(\tilde{\alpha}^{(L)}_m(x)\right) W^{(L)}_{mk}$$

First note that $\mathrm{Tr}\left(\nabla f_{\theta,m}(x)\left(\nabla f_{\theta,m}(x)\right)^T\right) = \Theta^{(L)}_{mm}(x,x)$. Now let $n_1, ... n_{L-1} \to \infty$, by the induction hypothesis, the pairs $(g^{(L)}_{\theta,m}, \tilde{\alpha}^{(L)}_m)$ converge to iid Gaussian pairs of processes with covariance $\Phi^{(L)}_\infty$ at initialization.

At initialization, conditioned on the values of $g^{(L)}_m, \tilde{\alpha}^{(L)}_m$ the pairs $(g^{(L+1)}_k, f_\theta)$ follow a centered Gaussian distribution with (conditioned) covariance

$$\mathbb{E}[g^{(L+1)}_{\theta,k}(x)g^{(L+1)}_{\theta,k'}(x')|g^{(L)}_{\theta,m}, \tilde{\alpha}^{(L)}_m] = \frac{\delta_{kk'}}{n_L} \sum_{m=1}^{n_L} \left(g^{(L)}_{\theta,m}(x)\dot{\sigma}\left(\tilde{\alpha}^{(L)}_m(x)\right) + \Theta^{(L)}_\infty(x,x)\ddot{\sigma}\left(\tilde{\alpha}^{(L)}_m(x)\right)\right)$$

$$\left(g^{(L)}_{\theta,m}(x')\dot{\sigma}\left(\tilde{\alpha}^{(L)}_m(x')\right) + \Theta^{(L)}_\infty(x',x')\ddot{\sigma}\left(\tilde{\alpha}^{(L)}_m(x')\right)\right)$$

$$\mathbb{E}[g^{(L+1)}_{\theta,k}(x)f_{\theta,k'}(x')|g^{(L)}_{\theta,m}, \tilde{\alpha}^{(L)}_m] = \frac{\delta_{kk'}}{n_L} \sum_{m=1}^{n_L} \left(g^{(L)}_{\theta,m}(x)\dot{\sigma}\left(\tilde{\alpha}^{(L)}_m(x)\right) + \Theta^{(L)}_\infty(x,x)\ddot{\sigma}\left(\tilde{\alpha}^{(L)}_m(x)\right)\right)$$

$$\sigma\left(\tilde{\alpha}^{(L)}_m(x')\right)$$

$$\mathbb{E}[f_{\theta,k}(x)f_{\theta,k'}(x')|g^{(L)}_{\theta,m}, \tilde{\alpha}^{(L)}_m] = \frac{\delta_{kk'}}{n_L} \sum_{m=1}^{n_L} \sigma\left(\tilde{\alpha}^{(L)}_m(x)\right)\sigma\left(\tilde{\alpha}^{(L)}_m(x')\right) + \beta^2.$$

As $n_L \to \infty$, by the law of large number, these (random) covariances converge to their expectations which are deterministic, hence the pairs $(g^{(L+1)}_k, f_{\theta k})$ have asymptotically the same Gaussian distribution independent of $g^{(L)}_m, \tilde{\alpha}^{(L)}_m$:

$$\mathbb{E}\left[g^{(L)}_{\theta,k}(x)g^{(L)}_{\theta,k'}(x')\right] \to \delta_{kk'}\Xi^{(L)}_\infty(x,x')$$

$$\mathbb{E}\left[g_{\theta,k}^{(L)}(x)f_{\theta,k'}^{(L)}(x')\right] \to \delta_{kk'}\Phi_\infty^{(L)}(x,x)$$

$$\mathbb{E}\left[f_{\theta,k}^{(L)}(x)f_{\theta,k'}^{(L)}(x')\right] \to \delta_{kk'}\Sigma_\infty^{(L)}(x,x)$$

with $\Xi_\infty^{(1)}(x,x') = \Phi_\infty^{(1)}(x,x') = 0$ and

$$\begin{aligned}
\Xi_\infty^{(L+1)}(x,x') &= \mathbb{E}\left[gg'\dot\sigma(\alpha)\dot\sigma(\alpha')\right] \\
&\quad + \Theta_\infty^{(L)}(x',x')\mathbb{E}\left[g\dot\sigma(\alpha)\ddot\sigma(\alpha')\right] \\
&\quad + \Theta_\infty^{(L)}(x,x)\mathbb{E}\left[g'\dot\sigma(\alpha')\ddot\sigma(\alpha)\right] \\
&\quad + \Theta_\infty^{(L)}(x,x)\Theta_\infty^{(L)}(x',x')\mathbb{E}\left[\ddot\sigma(\alpha')\ddot\sigma(\alpha)\right] \\
&= \Xi_\infty^{(L)}(x,x')\dot\Sigma_\infty^{(L)}(x,x') + \left(\Phi_\infty^{(L)}(x,x')\Phi_\infty^{(L)}(x',x) + \Phi_\infty^{(L)}(x,x)\Phi_\infty^{(L)}(x',x')\right)\ddot\Sigma_\infty^{(L)}(x,x') \\
&\quad + \Phi_\infty^{(L)}(x,x')\Phi_\infty^{(L)}(x',x')\mathbb{E}\left[\dot\sigma(\alpha)\ddot\sigma(\alpha')\right] + \Phi_\infty^{(L)}(x,x)\Phi_\infty^{(L)}(x',x)\mathbb{E}\left[\ddot\sigma(\alpha)\dot\sigma(\alpha')\right] \\
&\quad + \Theta_\infty^{(L)}(x',x')\left(\Phi_\infty^{(L)}(x,x)\ddot\Sigma_\infty^{(L)}(x,x') + \Phi_\infty^{(L)}(x,x')\mathbb{E}\left[\dot\sigma(\alpha)\ddot\sigma(\alpha')\right]\right) \\
&\quad + \Theta_\infty^{(L)}(x,x)\left(\Phi_\infty^{(L)}(x',x')\ddot\Sigma_\infty^{(L)}(x,x') + \Phi_\infty^{(L)}(x',x)\mathbb{E}\left[\ddot\sigma(\alpha)\dot\sigma(\alpha')\right]\right) \\
&\quad + \Theta_\infty^{(L)}(x,x)\Theta_\infty^{(L)}(x',x')\ddot\Sigma_\infty^{(L)}(x,x')
\end{aligned}$$

and

$$\begin{aligned}
\Phi_\infty^{(L+1)}(x,x') &= \mathbb{E}\left[g\dot\sigma(\alpha)\sigma(\alpha')\right] + \Theta_\infty^{(L)}(x,x)\mathbb{E}\left[\ddot\sigma(\alpha)\sigma(\alpha')\right] \\
&= \Phi_\infty^{(L)}(x,x')\dot\Sigma^{(L+1)}(x,x') + \left(\Phi_\infty^{(L)}(x,x) + \Theta_\infty^{(L)}(x,x)\right)\mathbb{E}\left[\ddot\sigma(\alpha)\sigma(\alpha')\right]
\end{aligned}$$

where $(g,g',\alpha,\alpha')$ is a Gaussian quadruple of covariance

$$\begin{pmatrix}
\Xi_\infty^{(L)}(x,x) & \Xi_\infty^{(L)}(x,x') & \Phi_\infty^{(L)}(x,x) & \Phi_\infty^{(L)}(x,x') \\
\Xi_\infty^{(L)}(x,x') & \Xi_\infty^{(L)}(x',x') & \Phi_\infty^{(L)}(x',x) & \Phi_\infty^{(L)}(x',x') \\
\Phi_\infty^{(L)}(x,x) & \Phi_\infty^{(L)}(x',x) & \Sigma_\infty^{(L)}(x,x) & \Sigma_\infty^{(L)}(x,x') \\
\Phi_\infty^{(L)}(x,x') & \Phi_\infty^{(L)}(x',x') & \Sigma_\infty^{(L)}(x,x') & \Sigma_\infty^{(L)}(x',x')
\end{pmatrix}.$$

During training, the parameters follow the gradient $\partial_t\theta(t) = (\partial_\theta Y(t))^T D(t)$. By the induction hypothesis, the traces $g_{\theta,m}^{(L)}$ then evolve according to the differential equation

$$\partial_t g_{\theta,m}^{(L)}(x) = \frac{1}{\sqrt{n_L}}\sum_{i=1}^N\sum_{m=1}^{n_L}\Lambda_{mm'}^{(L)}(x,x_i)\dot\sigma(\tilde\alpha_{m'}^{(L)}(x))\left(W_{m'}^{(L)}\right)^T D_i(t)$$

and in the limit as $n_1,...,n_{L-1}\to\infty$, the kernel $\Lambda_{mm'}^{(L)}(x,x_i)$ converges to a deterministic and fixed limit $\delta_{mm'}\Lambda_\infty^{(L)}(x,x_i)$. Note that as $n_L$ grows, the $g_{\theta,m}^{(L)}(x)$ move at a rate of $1/\sqrt{n_L}$ just like the pre-activations $\tilde\alpha_m^{(L)}$. Even though they move less and less, together they affect the trace $g_{\theta,k}^{(L+1)}$ which follows the differential equation

$$\partial_t g_{\theta,k}^{(L+1)}(x) = \sum_{i=1}^N\sum_{k'=1}^{n_L}\Lambda_{kk'}^{(L+1)}(x,x_i)D_{ik'}(t)$$

where

$$\begin{aligned}
\Lambda_{kk'}^{(L+1)}(x,x') &= \frac{1}{n_L}\sum_{m,m'}\Lambda_{mm'}^{(L)}(x,x')\dot\sigma\left(\tilde\alpha_m^{(L)}(x)\right)\dot\sigma\left(\tilde\alpha_{m'}^{(L)}(x')\right)W_{mk}^{(L)}W_{m'k'}^{(L)} \\
&\quad + \frac{1}{n_L}\sum_{m,m'}g_{\theta,m}^{(L)}(x)\Theta_{mm'}^{(L)}(x,x')\ddot\sigma\left(\tilde\alpha_m^{(L)}(x)\right)\dot\sigma\left(\tilde\alpha_{m'}^{(L)}(x')\right)W_{mk}^{(L)}W_{m'k'}^{(L)} \\
&\quad + \frac{1}{n_L}\sum_m g_{\theta,m}^{(L)}(x)\dot\sigma\left(\tilde\alpha_m^{(L)}(x)\right)\sigma\left(\tilde\alpha_m^{(L)}(x')\right)\delta_{kk'}
\end{aligned}$$

$$+ \frac{2}{n_L} \sum_{m,m'} \Omega^{(L)}_{m'mm}(x',x,x)\ddot{\sigma}\left(\tilde{\alpha}^{(L)}_m(x)\right)\dot{\sigma}\left(\tilde{\alpha}^{(L)}_{m'}(x')\right)W^{(L)}_{mk}W^{(L)}_{m'k'}$$

$$+ \frac{1}{n_L} \sum_{m,m'} \Theta^{(L)}_{mm}(x,x)\Theta^{(L)}_{mm'}(x,x')\ddot{\sigma}\left(\tilde{\alpha}^{(L)}_m(x)\right)\dot{\sigma}\left(\tilde{\alpha}^{(L)}_{m'}(x')\right)W^{(L)}_{mk}W^{(L)}_{m'k'}$$

$$+ \frac{1}{n_L} \sum_{m} \Theta^{(L)}_{mm}(x,x)\ddot{\sigma}\left(\tilde{\alpha}^{(L)}_m(x)\right)\sigma\left(\tilde{\alpha}^{(L)}_m(x')\right)\delta_{kk'}.$$

As $n_1,...,n_{L-1}\to\infty$, the kernels $\Theta^{(L)}_{mm'}(x,x')$ and $\Lambda^{(L)}_{mm'}(x,x')$ converge to their limit and $\Omega^{(L)}_{m'mm}(x',x,x)$ vanishes:

$$\Lambda^{(L)}_{kk'}(x,x') \to \frac{1}{n_L}\sum_{m}\Lambda^{(L)}_\infty(x,x')\dot{\sigma}\left(\tilde{\alpha}^{(L)}_m(x)\right)\dot{\sigma}\left(\tilde{\alpha}^{(L)}_m(x')\right)W^{(L)}_{mk}W^{(L)}_{mk'}$$

$$+ \frac{1}{n_L}\sum_{m}g^{(L)}_{\theta,m}(x)\Theta^{(L)}_\infty(x,x')\ddot{\sigma}\left(\tilde{\alpha}^{(L)}_m(x)\right)\dot{\sigma}\left(\tilde{\alpha}^{(L)}_m(x')\right)W^{(L)}_{mk}W^{(L)}_{mk'}$$

$$+ \frac{1}{n_L}\sum_{m}g^{(L)}_{\theta,m}(x)\dot{\sigma}\left(\tilde{\alpha}^{(L)}_m(x)\right)\sigma\left(\tilde{\alpha}^{(L)}_m(x')\right)\delta_{kk'}$$

$$+ \frac{1}{n_L}\sum_{m}\Theta^{(L)}_\infty(x,x)\Theta^{(L)}_\infty(x,x')\ddot{\sigma}\left(\tilde{\alpha}^{(L)}_m(x)\right)\dot{\sigma}\left(\tilde{\alpha}^{(L)}_m(x')\right)W^{(L)}_{mk}W^{(L)}_{mk'}$$

$$+ \frac{1}{n_L}\sum_{m}\Theta^{(L)}_\infty(x,x)\ddot{\sigma}\left(\tilde{\alpha}^{(L)}_m(x)\right)\sigma\left(\tilde{\alpha}^{(L)}_m(x')\right)\delta_{kk'}$$

By the law of large numbers, as $n_L\to\infty$, at initialization $\Lambda^{(L+1)}_{kk'}(x,x')\to\delta_{kk'}\Lambda^{(L+1)}_\infty(x,x')$ where

$$\Lambda^{(L+1)}_\infty(x,x') = \Lambda^{(L)}_\infty(x,x')\dot{\Sigma}^{(L+1)}_\infty(x,x')$$
$$+ \Theta^{(L)}_\infty(x,x')\mathbb{E}\left[g\ddot{\sigma}(\alpha)\dot{\sigma}(\alpha')\right]$$
$$+ \mathbb{E}\left[g\dot{\sigma}(\alpha)\sigma(\alpha')\right]$$
$$+ \Theta^{(L)}_\infty(x,x)\Theta^{(L)}_\infty(x,x')\mathbb{E}\left[\ddot{\sigma}(\alpha)\dot{\sigma}(\alpha')\right]$$
$$+ \Theta^{(L)}_\infty(x,x)\mathbb{E}\left[\ddot{\sigma}(\alpha)\sigma(\alpha')\right]$$
$$= \Lambda^{(L)}_\infty(x,x')\dot{\Sigma}^{(L+1)}_\infty(x,x')$$
$$+ \Theta^{(L)}_\infty(x,x')\left(\Phi^{(L)}_\infty(x,x')\ddot{\Sigma}^{(L+1)}_\infty(x,x') + \Phi^{(L)}_\infty(x,x)\mathbb{E}\left[\ddot{\sigma}(\alpha)\dot{\sigma}(\alpha')\right]\right)$$
$$+ \Phi^{(L)}_\infty(x,x')\dot{\Sigma}^{(L+1)}_\infty(x,x') + \Phi^{(L)}_\infty(x,x)\mathbb{E}\left[\ddot{\sigma}(\alpha)\sigma(\alpha')\right]$$
$$+ \Theta^{(L)}_\infty(x,x)\Theta^{(L)}_\infty(x,x')\mathbb{E}\left[\ddot{\sigma}(\alpha)\dot{\sigma}(\alpha')\right]$$
$$+ \Theta^{(L)}_\infty(x,x)\mathbb{E}\left[\ddot{\sigma}(\alpha)\dot{\sigma}(\alpha')\right]$$

During training $\Theta^{(L)}_\infty$ and $\Lambda^{(L)}_\infty$ are fixed in the limit $n_1,..,n_{L-1}\to\infty$, and the values $g^{(L)}_{\theta,m}(x)$, $\tilde{\alpha}^{(L)}_m(x)$ and $W^{(L)}_{mk}$ vary at a rate of $1/\sqrt{n_L}$ which induce a change of the same rate to $\Lambda^{(L)}_{kk'}(x,x')$, which is therefore asymptotically fixed during training as $n_L\to\infty$. $\qquad\square$

The next lemma describes the asymptotic limit of the kernel $\Upsilon^{(L)}$:

**Lemma 4.** *For any loss $C$ with BGOSS and $\sigma\in C^4_b(\mathbb{R})$, the second moment of the Hessian of the realization function $\mathcal{H}F^{(L)}$ converges uniformly over $[0,T]$ to a fixed limit as $n_1,...n_{L-1}\to\infty$*

$$\Upsilon^{(L)}_{kk'}(x,x') \to \delta_{kk'}\sum_{\ell=1}^{L-1}\left(\Theta^{(\ell)}_\infty(x,x')^2\ddot{\Sigma}^{(\ell)}_\infty(x,x') + 2\Theta^{(\ell)}_\infty(x,x')\dot{\Sigma}^{(\ell)}_\infty(x,x')\right)\dot{\Sigma}^{(\ell+1)}_\infty(x,x')\cdots\dot{\Sigma}^{(L-1)}_\infty(x,x').$$

*Proof.* The proof is by induction on the depth $L$. The case $L = 1$ is trivially true because $\partial^2_{\theta_p \theta_{p'}} f_{\theta,k}(x) = 0$ for all $p, p', k, x$. For the induction step we observe that

$$\Upsilon^{(L)}_{k,k'}(x, x')$$

$$= \sum_{p_1, p_2 = 1}^{P} \partial^2_{\theta_{p_1}, \theta_{p_2}} f_{\theta,k}(x) \partial^2_{\theta_{p_2}, \theta_{p_1}} f_{\theta,k'}(x')$$

$$= \frac{1}{n_L} \sum_{m,m'=1}^{n_L} \Upsilon^{(L)}_{m,m'}(x, x') \dot{\sigma}\left(\tilde{\alpha}^{(L)}_m(x)\right) \dot{\sigma}\left(\tilde{\alpha}^{(L)}_{m'}(x')\right) W^{(L)}_{mk} W^{(L)}_{m'k'}$$

$$+ \frac{1}{n_L} \sum_{m,m'=1}^{n_L} \Omega^{(L)}_{m',m,m'}(x', x, x') \dot{\sigma}\left(\tilde{\alpha}^{(L)}_m(x)\right) \ddot{\sigma}\left(\tilde{\alpha}^{(L)}_{m'}(x')\right) W^{(L)}_{mk} W^{(L)}_{m'k'}$$

$$+ \frac{1}{n_L} \sum_{m,m'=1}^{n_L} \Omega^{(L)}_{m,m',m}(x, x', x) \ddot{\sigma}\left(\tilde{\alpha}^{(L)}_m(x)\right) \dot{\sigma}\left(\tilde{\alpha}^{(L)}_{m'}(x')\right) W^{(L)}_{mk} W^{(L)}_{m'k'}$$

$$+ \frac{1}{n_L} \sum_{m,m'=1}^{n_L} \Theta^{(L)}_{m,m'}(x, x') \Theta^{(L)}_{m',m}(x', x) \ddot{\sigma}\left(\tilde{\alpha}^{(L)}_m(x)\right) \ddot{\sigma}\left(\tilde{\alpha}^{(L)}_{m'}(x')\right) W^{(L)}_{mk} W^{(L)}_{m'k'}$$

$$+ \frac{2}{n_L} \sum_{m=1}^{n_L} \Theta^{(L)}_{m,m'}(x, x') \dot{\sigma}\left(\tilde{\alpha}^{(L)}_m(x)\right) \dot{\sigma}\left(\tilde{\alpha}^{(L)}_{m'}(x')\right) \delta_{kk'}$$

if we now let the width of the lower layers grow to infinity $n_1, ... n_{L-1} \to \infty$, the tensor $\Omega^{(L)}$ vanishes and $\Upsilon^{(L)}_{m,m'}$ and the NTK $\Theta^{(L)}_{m,m'}$ converge to limits which are non-zero only when $m = m'$. As a result, the term above converges to

$$\frac{1}{n_L} \sum_{m=1}^{n_L} \Upsilon^{(L)}_{\infty}(x, x') \dot{\sigma}\left(\tilde{\alpha}^{(L)}_m(x)\right) \dot{\sigma}\left(\tilde{\alpha}^{(L)}_m(x')\right) W^{(L)}_{mk} W^{(L)}_{mk'}$$

$$+ \frac{1}{n_L} \sum_{m=1}^{n_L} \Theta^{(L)}_{\infty}(x, x')^2 \ddot{\sigma}\left(\tilde{\alpha}^{(L)}_m(x)\right) \ddot{\sigma}\left(\tilde{\alpha}^{(L)}_m(x')\right) W^{(L)}_{mk} W^{(L)}_{mk'}$$

$$+ \frac{2}{n_L} \sum_{m=1}^{n_L} \Theta^{(L)}_{\infty}(x, x') \dot{\sigma}\left(\tilde{\alpha}^{(L)}_m(x)\right) \dot{\sigma}\left(\tilde{\alpha}^{(L)}_m(x')\right) \delta_{kk'}$$

At initialization, we can apply the law of large numbers as $n_L \to \infty$ such that it converges to $\Upsilon^{(L+1)}_{\infty}(x, x') \delta_{kk'}$, for the kernel $\Upsilon^{(L+1)}_{\infty}(x, x')$ defined recursively by

$$\Upsilon^{(L+1)}_{\infty}(x, x') = \Upsilon^{(L)}_{\infty}(x, x') \dot{\Sigma}^{(L)}_{\infty}(x, x') + \Theta^{(L)}_{\infty}(x, x')^2 \ddot{\Sigma}^{(L)}_{\infty}(x, x') + 2\Theta^{(L)}_{\infty}(x, x') \dot{\Sigma}^{(L)}_{\infty}(x, x')$$

and $\Upsilon^{(1)}_{\infty}(x, x') = 0$.

For the convergence during training, we proceed similarily to the proof of Lemma 1: the activations $\tilde{\alpha}^{(L)}_m(x)$ and weights $W^{(L)}_{mk}$ move at a rate of $1/\sqrt{n_L}$ and the change to $\Upsilon^{(L+1)}_{kk'}$ is therefore of order $1/\sqrt{n_L}$ and vanishes as $n_L \to 0$. $\qquad\square$

Finally, the next lemma shows the vanishing of the tensor $\Psi^{(L)}_{k_0,k_1,k_2,k_3}$ to prove that the higher moments of $S$ vanish.

**Lemma 5.** *For any loss $C$ with BGOSS and $\sigma \in C^4_b(\mathbb{R})$, uniformly over $[0, T]$*

$$\lim_{n_{L-1} \to \infty} \cdots \lim_{n_1 \to \infty} \Psi^{(L)}_{k_0,k_1,k_2,k_3}(x_{i_0}, x_{i_1}, x_{i_2}, x_{i_3}) = 0$$

*Proof.* When $L = 1$ the Hessian is zero and $\Psi^{(1)}_{k_0,k_1,k_2,k_3}(x_{i_0}, x_{i_1}, x_{i_2}, x_{i_3}) = 0$.

For the induction step, we write $\Psi^{(L+1)}_{k_0,k_1,k_2,k_3}(x_{i_0}, x_{i_1}, x_{i_2}, x_{i_3})$ recursively, because it contains many terms, we change the notation, writing $\begin{bmatrix} x_0 & x_1 \\ m_0 & m_1 \end{bmatrix}$ for

$\Theta^{(L)}_{m_0,m_1}(x_0,x_1)$, $\begin{bmatrix} x_0 & x_1 & x_2 \\ m_0 & m_1 & m_2 \end{bmatrix}$ for $\Omega^{(L)}_{m_0,m_1,m_2}(x_0,x_1,x_2)$ and $\begin{bmatrix} x_0 & x_1 & x_2 & x_3 \\ m_0 & m_1 & m_2 & m_3 \end{bmatrix}$ for $\Gamma^{(L)}_{m_0,m_1,m_2,m_3}(x_0,x_1,x_2,x_3)$. The value $\Psi^{(L+1)}_{k_0,k_1,k_2,k_3}(x_{i_0},x_{i_1},x_{i_2},x_{i_3})$ is then equal to

$$n_L^{-2}\sum_{m_0,m_1,m_2,m_3}\Psi^{(L)}_{m_0,m_1,m_2,m_3}(x_0,x_1,x_2,x_3)\dot{\sigma}\left(\tilde{\alpha}^{(L)}_{m_0}(x_0)\right)\dot{\sigma}\left(\tilde{\alpha}^{(L)}_{m_1}(x_1)\right)\dot{\sigma}\left(\tilde{\alpha}^{(L)}_{m_2}(x_2)\right)$$

$$\dot{\sigma}\left(\tilde{\alpha}^{(L)}_{m_3}(x_3)\right)W^{(L)}_{m_0k_0}W^{(L)}_{m_1k_1}W^{(L)}_{m_2k_2}W^{(L)}_{m_3k_3}$$

$$+n_L^{-2}\sum_{m_0,m_1,m_2,m_3}\begin{bmatrix} x_0 & x_1 \\ m_0 & m_1 \end{bmatrix}\begin{bmatrix} x_1 & x_2 \\ m_1 & m_2 \end{bmatrix}\begin{bmatrix} x_2 & x_3 \\ m_2 & m_3 \end{bmatrix}\begin{bmatrix} x_3 & x_0 \\ m_3 & m_0 \end{bmatrix}\ddot{\sigma}\left(\tilde{\alpha}^{(L)}_{m_0}(x_0)\right)$$

$$\ddot{\sigma}\left(\tilde{\alpha}^{(L)}_{m_1}(x_1)\right)\ddot{\sigma}\left(\tilde{\alpha}^{(L)}_{m_2}(x_2)\right)\ddot{\sigma}\left(\tilde{\alpha}^{(L)}_{m_3}(x_3)\right)W^{(L)}_{m_0k_0}W^{(L)}_{m_1k_1}W^{(L)}_{m_2k_2}W^{(L)}_{m_3k_3}$$

$$+n_L^{-2}\sum_{m_0,m_1,m_2,m_3}\begin{bmatrix} x_0 & x_1 & x_2 \\ m_0 & m_1 & m_2 \end{bmatrix}\begin{bmatrix} x_2 & x_3 \\ m_2 & m_3 \end{bmatrix}\begin{bmatrix} x_3 & x_0 \\ m_3 & m_0 \end{bmatrix}\ddot{\sigma}\left(\tilde{\alpha}^{(L)}_{m_0}(x_0)\right)\dot{\sigma}\left(\tilde{\alpha}^{(L)}_{m_1}(x_1)\right)$$

$$\ddot{\sigma}\left(\tilde{\alpha}^{(L)}_{m_2}(x_2)\right)\ddot{\sigma}\left(\tilde{\alpha}^{(L)}_{m_3}(x_3)\right)W^{(L)}_{m_0k_0}W^{(L)}_{m_1k_1}W^{(L)}_{m_2k_2}W^{(L)}_{m_3k_3}$$

$$+n_L^{-2}\sum_{m_0,m_1,m_2,m_3}\begin{bmatrix} x_0 & x_1 \\ m_0 & m_1 \end{bmatrix}\begin{bmatrix} x_1 & x_2 & x_3 \\ m_1 & m_2 & m_3 \end{bmatrix}\begin{bmatrix} x_3 & x_0 \\ m_3 & m_0 \end{bmatrix}\ddot{\sigma}\left(\tilde{\alpha}^{(L)}_{m_0}(x_0)\right)\ddot{\sigma}\left(\tilde{\alpha}^{(L)}_{m_1}(x_1)\right)$$

$$\dot{\sigma}\left(\tilde{\alpha}^{(L)}_{m_2}(x_2)\right)\ddot{\sigma}\left(\tilde{\alpha}^{(L)}_{m_3}(x_3)\right)W^{(L)}_{m_0k_0}W^{(L)}_{m_1k_1}W^{(L)}_{m_2k_2}W^{(L)}_{m_3k_3}$$

$$+n_L^{-2}\sum_{m_0,m_1,m_2,m_3}\begin{bmatrix} x_0 & x_1 \\ m_0 & m_1 \end{bmatrix}\begin{bmatrix} x_1 & x_2 \\ m_1 & m_2 \end{bmatrix}\begin{bmatrix} x_2 & x_3 & x_0 \\ m_2 & m_3 & m_0 \end{bmatrix}\ddot{\sigma}\left(\tilde{\alpha}^{(L)}_{m_0}(x_0)\right)\ddot{\sigma}\left(\tilde{\alpha}^{(L)}_{m_1}(x_1)\right)$$

$$\ddot{\sigma}\left(\tilde{\alpha}^{(L)}_{m_2}(x_2)\right)\dot{\sigma}\left(\tilde{\alpha}^{(L)}_{m_3}(x_3)\right)W^{(L)}_{m_0k_0}W^{(L)}_{m_1k_1}W^{(L)}_{m_2k_2}W^{(L)}_{m_3k_3}$$

$$+n_L^{-2}\sum_{m_0,m_1,m_2,m_3}\begin{bmatrix} x_1 & x_2 \\ m_1 & m_2 \end{bmatrix}\begin{bmatrix} x_2 & x_3 \\ m_2 & m_3 \end{bmatrix}\begin{bmatrix} x_3 & x_0 & x_1 \\ m_3 & m_0 & m_1 \end{bmatrix}\dot{\sigma}\left(\tilde{\alpha}^{(L)}_{m_0}(x_0)\right)\ddot{\sigma}\left(\tilde{\alpha}^{(L)}_{m_1}(x_1)\right)$$

$$\ddot{\sigma}\left(\tilde{\alpha}^{(L)}_{m_2}(x_2)\right)\ddot{\sigma}\left(\tilde{\alpha}^{(L)}_{m_3}(x_3)\right)W^{(L)}_{m_0k_0}W^{(L)}_{m_1k_1}W^{(L)}_{m_2k_2}W^{(L)}_{m_3k_3}$$

$$+n_L^{-2}\sum_{m_0,m_1,m_2,m_3}\begin{bmatrix} x_0 & x_1 & x_2 \\ m_0 & m_1 & m_2 \end{bmatrix}\begin{bmatrix} x_2 & x_3 & x_0 \\ m_2 & m_3 & m_0 \end{bmatrix}\ddot{\sigma}\left(\tilde{\alpha}^{(L)}_{m_0}(x_0)\right)\dot{\sigma}\left(\tilde{\alpha}^{(L)}_{m_1}(x_1)\right)$$

$$\ddot{\sigma}\left(\tilde{\alpha}^{(L)}_{m_2}(x_2)\right)\dot{\sigma}\left(\tilde{\alpha}^{(L)}_{m_3}(x_3)\right)W^{(L)}_{m_0k_0}W^{(L)}_{m_1k_1}W^{(L)}_{m_2k_2}W^{(L)}_{m_3k_3}$$

$$+n_L^{-2}\sum_{m_0,m_1,m_2,m_3}\begin{bmatrix} x_1 & x_2 & x_3 \\ m_1 & m_2 & m_3 \end{bmatrix}\begin{bmatrix} x_3 & x_0 & x_1 \\ m_3 & m_0 & m_1 \end{bmatrix}\dot{\sigma}\left(\tilde{\alpha}^{(L)}_{m_0}(x_0)\right)\ddot{\sigma}\left(\tilde{\alpha}^{(L)}_{m_1}(x_1)\right)$$

$$\dot{\sigma}\left(\tilde{\alpha}^{(L)}_{m_2}(x_2)\right)\ddot{\sigma}\left(\tilde{\alpha}^{(L)}_{m_3}(x_3)\right)W^{(L)}_{m_0k_0}W^{(L)}_{m_1k_1}W^{(L)}_{m_2k_2}W^{(L)}_{m_3k_3}$$

$$+n_L^{-2}\sum_{m_0,m_1,m_2,m_3}\begin{bmatrix} x_0 & x_1 & x_2 & x_3 \\ m_0 & m_1 & m_2 & m_3 \end{bmatrix}\begin{bmatrix} x_3 & x_0 \\ m_3 & m_0 \end{bmatrix}\ddot{\sigma}\left(\tilde{\alpha}^{(L)}_{m_0}(x_0)\right)\dot{\sigma}\left(\tilde{\alpha}^{(L)}_{m_1}(x_1)\right)$$

$$\dot{\sigma}\left(\tilde{\alpha}^{(L)}_{m_2}(x_2)\right)\ddot{\sigma}\left(\tilde{\alpha}^{(L)}_{m_3}(x_3)\right)W^{(L)}_{m_0k_0}W^{(L)}_{m_1k_1}W^{(L)}_{m_2k_2}W^{(L)}_{m_3k_3}$$

$$+n_L^{-2}\sum_{m_0,m_1,m_2,m_3}\begin{bmatrix} x_0 & x_1 \\ m_0 & m_1 \end{bmatrix}\begin{bmatrix} x_1 & x_2 & x_3 & x_0 \\ m_1 & m_2 & m_3 & m_0 \end{bmatrix}\ddot{\sigma}\left(\tilde{\alpha}^{(L)}_{m_0}(x_0)\right)\ddot{\sigma}\left(\tilde{\alpha}^{(L)}_{m_1}(x_1)\right)$$

$$\dot{\sigma}\left(\tilde{\alpha}^{(L)}_{m_2}(x_2)\right)\dot{\sigma}\left(\tilde{\alpha}^{(L)}_{m_3}(x_3)\right)W^{(L)}_{m_0k_0}W^{(L)}_{m_1k_1}W^{(L)}_{m_2k_2}W^{(L)}_{m_3k_3}$$

$$+n_L^{-2}\sum_{m_0,m_1,m_2,m_3}\begin{bmatrix} x_1 & x_2 \\ m_1 & m_2 \end{bmatrix}\begin{bmatrix} x_2 & x_3 & x_0 & x_1 \\ m_2 & m_3 & m_0 & m_1 \end{bmatrix}\dot{\sigma}\left(\tilde{\alpha}^{(L)}_{m_0}(x_0)\right)\ddot{\sigma}\left(\tilde{\alpha}^{(L)}_{m_1}(x_1)\right)$$

$$\ddot{\sigma}\left(\tilde{\alpha}^{(L)}_{m_2}(x_2)\right)\dot{\sigma}\left(\tilde{\alpha}^{(L)}_{m_3}(x_3)\right)W^{(L)}_{m_0k_0}W^{(L)}_{m_1k_1}W^{(L)}_{m_2k_2}W^{(L)}_{m_3k_3}$$

$$
+n_L^{-2} \sum_{m_0,m_1,m_2,m_3} \begin{bmatrix} x_2 & x_3 \\ m_2 & m_3 \end{bmatrix} \begin{bmatrix} x_3 & x_0 & x_1 & x_2 \\ m_3 & m_0 & m_1 & m_2 \end{bmatrix} \dot{\sigma}\left(\tilde{\alpha}_{m_0}^{(L)}(x_0)\right) \dot{\sigma}\left(\tilde{\alpha}_{m_1}^{(L)}(x_1)\right)
$$

$$
\ddot{\sigma}\left(\tilde{\alpha}_{m_2}^{(L)}(x_2)\right) \ddot{\sigma}\left(\tilde{\alpha}_{m_3}^{(L)}(x_3)\right) W_{m_0 k_0}^{(L)} W_{m_1 k_1}^{(L)} W_{m_2 k_2}^{(L)} W_{m_3 k_3}^{(L)}
$$

$$
+n_L^{-2} \sum_{m,m_1,m_2} \begin{bmatrix} x_0 & x_1 \\ m & m_1 \end{bmatrix} \begin{bmatrix} x_1 & x_2 \\ m_1 & m_2 \end{bmatrix} \begin{bmatrix} x_2 & x_3 \\ m_2 & m \end{bmatrix} \dot{\sigma}\left(\tilde{\alpha}_{m}^{(L)}(x_0)\right) \ddot{\sigma}\left(\tilde{\alpha}_{m_1}^{(L)}(x_1)\right)
$$

$$
\ddot{\sigma}\left(\tilde{\alpha}_{m_2}^{(L)}(x_2)\right) \dot{\sigma}\left(\tilde{\alpha}_{m}^{(L)}(x_3)\right) W_{m_1 k_1}^{(L)} W_{m_2 k_2}^{(L)} \delta_{k_0 k_3}
$$

$$
+n_L^{-2} \sum_{m,m_2,m_3} \begin{bmatrix} x_1 & x_2 \\ m & m_2 \end{bmatrix} \begin{bmatrix} x_2 & x_3 \\ m_2 & m_3 \end{bmatrix} \begin{bmatrix} x_3 & x_0 \\ m_3 & m \end{bmatrix} \dot{\sigma}\left(\tilde{\alpha}_{m}^{(L)}(x_0)\right) \dot{\sigma}\left(\tilde{\alpha}_{m}^{(L)}(x_1)\right)
$$

$$
\ddot{\sigma}\left(\tilde{\alpha}_{m_2}^{(L)}(x_2)\right) \ddot{\sigma}\left(\tilde{\alpha}_{m_3}^{(L)}(x_3)\right) W_{m_2 k_2}^{(L)} W_{m_3 k_3}^{(L)} \delta_{k_0 k_1}
$$

$$
+n_L^{-2} \sum_{m,m_3,m_0} \begin{bmatrix} x_0 & x_1 \\ m_0 & m \end{bmatrix} \begin{bmatrix} x_2 & x_3 \\ m & m_3 \end{bmatrix} \begin{bmatrix} x_3 & x_0 \\ m_3 & m_0 \end{bmatrix} \ddot{\sigma}\left(\tilde{\alpha}_{m_0}^{(L)}(x_0)\right) \dot{\sigma}\left(\tilde{\alpha}_{m}^{(L)}(x_1)\right)
$$

$$
\dot{\sigma}\left(\tilde{\alpha}_{m}^{(L)}(x_2)\right) \ddot{\sigma}\left(\tilde{\alpha}_{m_3}^{(L)}(x_3)\right) W_{m_0 k_0}^{(L)} W_{m_3 k_3}^{(L)} \delta_{k_1 k_2}
$$

$$
+n_L^{-2} \sum_{m,m_0,m_1} \begin{bmatrix} x_0 & x_1 \\ m_0 & m_1 \end{bmatrix} \begin{bmatrix} x_1 & x_2 \\ m_1 & m \end{bmatrix} \begin{bmatrix} x_3 & x_0 \\ m & m_0 \end{bmatrix} \ddot{\sigma}\left(\tilde{\alpha}_{m_0}^{(L)}(x_0)\right) \ddot{\sigma}\left(\tilde{\alpha}_{m_1}^{(L)}(x_1)\right)
$$

$$
\dot{\sigma}\left(\tilde{\alpha}_{m}^{(L)}(x_2)\right) \dot{\sigma}\left(\tilde{\alpha}_{m}^{(L)}(x_3)\right) W_{m_0 k_0}^{(L)} W_{m_1 k_1}^{(L)} \delta_{k_2 k_3}
$$

$$
+n_L^{-2} \sum_{m,m_1,m_2} \begin{bmatrix} x_0 & x_1 & x_2 \\ m & m_1 & m_2 \end{bmatrix} \begin{bmatrix} x_2 & x_3 \\ m_2 & m \end{bmatrix} \dot{\sigma}\left(\tilde{\alpha}_{m}^{(L)}(x_0)\right) \dot{\sigma}\left(\tilde{\alpha}_{m_1}^{(L)}(x_1)\right)
$$

$$
\ddot{\sigma}\left(\tilde{\alpha}_{m_2}^{(L)}(x_2)\right) \dot{\sigma}\left(\tilde{\alpha}_{m}^{(L)}(x_3)\right) W_{m_1 k_1}^{(L)} W_{m_2 k_2}^{(L)} \delta_{k_0 k_3}
$$

$$
+n_L^{-2} \sum_{m,m_2,m_3} \begin{bmatrix} x_1 & x_2 & x_3 \\ m & m_2 & m_3 \end{bmatrix} \begin{bmatrix} x_3 & x_0 \\ m_3 & m \end{bmatrix} \dot{\sigma}\left(\tilde{\alpha}_{m}^{(L)}(x_0)\right) \dot{\sigma}\left(\tilde{\alpha}_{m}^{(L)}(x_1)\right) \dot{\sigma}\left(\tilde{\alpha}_{m_2}^{(L)}(x_2)\right)
$$

$$
\ddot{\sigma}\left(\tilde{\alpha}_{m_3}^{(L)}(x_3)\right) W_{m_2 k_2}^{(L)} W_{m_3 k_3}^{(L)} \delta_{k_0 k_1}
$$

$$
+n_L^{-2} \sum_{m,m_3,m_0} \begin{bmatrix} x_0 & x_1 \\ m_0 & m \end{bmatrix} \begin{bmatrix} x_2 & x_3 & x_0 \\ m & m_3 & m_0 \end{bmatrix} \ddot{\sigma}\left(\tilde{\alpha}_{m_0}^{(L)}(x_0)\right) \dot{\sigma}\left(\tilde{\alpha}_{m}^{(L)}(x_1)\right)
$$

$$
\dot{\sigma}\left(\tilde{\alpha}_{m}^{(L)}(x_2)\right) \dot{\sigma}\left(\tilde{\alpha}_{m_3}^{(L)}(x_3)\right) W_{m_0 k_0}^{(L)} W_{m_3 k_3}^{(L)} \delta_{k_1 k_2}
$$

$$
+n_L^{-2} \sum_{m,m_0,m_1} \begin{bmatrix} x_1 & x_2 \\ m_1 & m \end{bmatrix} \begin{bmatrix} x_3 & x_0 & x_1 \\ m & m_0 & m_1 \end{bmatrix} \dot{\sigma}\left(\tilde{\alpha}_{m_0}^{(L)}(x_0)\right) \ddot{\sigma}\left(\tilde{\alpha}_{m_1}^{(L)}(x_1)\right)
$$

$$
\dot{\sigma}\left(\tilde{\alpha}_{m}^{(L)}(x_2)\right) \dot{\sigma}\left(\tilde{\alpha}_{m}^{(L)}(x_3)\right) W_{m_0 k_0}^{(L)} W_{m_1 k_1}^{(L)} \delta_{k_2 k_3}
$$

$$
+n_L^{-2} \sum_{m,m_1,m_2} \begin{bmatrix} x_0 & x_1 \\ m & m_1 \end{bmatrix} \begin{bmatrix} x_1 & x_2 & x_3 \\ m_1 & m_2 & m \end{bmatrix} \dot{\sigma}\left(\tilde{\alpha}_{m}^{(L)}(x_0)\right) \ddot{\sigma}\left(\tilde{\alpha}_{m_1}^{(L)}(x_1)\right)
$$

$$
\dot{\sigma}\left(\tilde{\alpha}_{m_2}^{(L)}(x_2)\right) \dot{\sigma}\left(\tilde{\alpha}_{m}^{(L)}(x_3)\right) W_{m_1 k_1}^{(L)} W_{m_2 k_2}^{(L)} \delta_{k_0 k_3}
$$

$$
+n_L^{-2} \sum_{m,m_2,m_3} \begin{bmatrix} x_1 & x_2 \\ m & m_2 \end{bmatrix} \begin{bmatrix} x_2 & x_3 & x_0 \\ m_2 & m_3 & m \end{bmatrix} \dot{\sigma}\left(\tilde{\alpha}_{m}^{(L)}(x_0)\right) \dot{\sigma}\left(\tilde{\alpha}_{m}^{(L)}(x_1)\right)
$$

$$
\ddot{\sigma}\left(\tilde{\alpha}_{m_2}^{(L)}(x_2)\right) \dot{\sigma}\left(\tilde{\alpha}_{m_3}^{(L)}(x_3)\right) W_{m_2 k_2}^{(L)} W_{m_3 k_3}^{(L)} \delta_{k_0 k_1}
$$

$$
+n_L^{-2} \sum_{m,m_3,m_0} \begin{bmatrix} x_2 & x_3 \\ m & m_3 \end{bmatrix} \begin{bmatrix} x_3 & x_0 & x_1 \\ m_3 & m_0 & m \end{bmatrix} \dot{\sigma}\left(\tilde{\alpha}_{m_0}^{(L)}(x_0)\right) \dot{\sigma}\left(\tilde{\alpha}_{m}^{(L)}(x_1)\right)
$$

$$
\dot{\sigma}\left(\tilde{\alpha}_{m}^{(L)}(x_2)\right) \ddot{\sigma}\left(\tilde{\alpha}_{m_3}^{(L)}(x_3)\right) W_{m_0 k_0}^{(L)} W_{m_3 k_3}^{(L)} \delta_{k_1 k_2}
$$

$$+n_L^{-2} \sum_{m,m_0,m_1} \left[ \begin{array}{ccc} x_0 & x_1 & x_2 \\ m_0 & m_1 & m \end{array} \right] \left[ \begin{array}{cc} x_3 & x_0 \\ m & m_0 \end{array} \right] \ddot{\sigma}\left(\tilde{\alpha}_{m_0}^{(L)}(x_0)\right) \dot{\sigma}\left(\tilde{\alpha}_{m_1}^{(L)}(x_1)\right)$$

$$\dot{\sigma}\left(\tilde{\alpha}_m^{(L)}(x_2)\right) \dot{\sigma}\left(\tilde{\alpha}_m^{(L)}(x_3)\right) W_{m_0 k_0}^{(L)} W_{m_1 k_1}^{(L)} \delta_{k_2 k_3}$$

$$+n_L^{-2} \sum_{m,m_1,m_2} \left[ \begin{array}{cccc} x_0 & x_1 & x_2 & x_3 \\ m & m_1 & m_2 & m \end{array} \right] \dot{\sigma}\left(\tilde{\alpha}_m^{(L)}(x_0)\right) \dot{\sigma}\left(\tilde{\alpha}_{m_1}^{(L)}(x_1)\right) \dot{\sigma}\left(\tilde{\alpha}_{m_2}^{(L)}(x_2)\right) \dot{\sigma}\left(\tilde{\alpha}_m^{(L)}(x_3)\right)$$

$$W_{m_1 k_1}^{(L)} W_{m_2 k_2}^{(L)} \delta_{k_0 k_3}$$

$$+n_L^{-2} \sum_{m,m_2,m_3} \left[ \begin{array}{cccc} x_1 & x_2 & x_3 & x_0 \\ m & m_2 & m_3 & m \end{array} \right] \dot{\sigma}\left(\tilde{\alpha}_m^{(L)}(x_0)\right) \dot{\sigma}\left(\tilde{\alpha}_m^{(L)}(x_1)\right) \dot{\sigma}\left(\tilde{\alpha}_{m_2}^{(L)}(x_2)\right) \dot{\sigma}\left(\tilde{\alpha}_{m_3}^{(L)}(x_3)\right)$$

$$W_{m_2 k_2}^{(L)} W_{m_3 k_3}^{(L)} \delta_{k_0 k_1}$$

$$+n_L^{-2} \sum_{m,m_3,m_0} \left[ \begin{array}{cccc} x_2 & x_3 & x_0 & x_1 \\ m & m_3 & m_0 & m \end{array} \right] \dot{\sigma}\left(\tilde{\alpha}_{m_0}^{(L)}(x_0)\right) \dot{\sigma}\left(\tilde{\alpha}_m^{(L)}(x_1)\right) \dot{\sigma}\left(\tilde{\alpha}_m^{(L)}(x_2)\right) \dot{\sigma}\left(\tilde{\alpha}_{m_3}^{(L)}(x_3)\right)$$

$$W_{m_0 k_0}^{(L)} W_{m_3 k_3}^{(L)} \delta_{k_1 k_2}$$

$$+n_L^{-2} \sum_{m,m_0,m_1} \left[ \begin{array}{cccc} x_3 & x_0 & x_1 & x_2 \\ m & m_0 & m_1 & m \end{array} \right] \dot{\sigma}\left(\tilde{\alpha}_{m_0}^{(L)}(x_0)\right) \dot{\sigma}\left(\tilde{\alpha}_{m_1}^{(L)}(x_1)\right) \dot{\sigma}\left(\tilde{\alpha}_m^{(L)}(x_2)\right) \dot{\sigma}\left(\tilde{\alpha}_m^{(L)}(x_3)\right)$$

$$W_{m_0 k_0}^{(L)} W_{m_1 k_1}^{(L)} \delta_{k_2 k_3}$$

$$+n_L^{-2} \sum_{m,m'} \left[ \begin{array}{cc} x_0 & x_1 \\ m & m' \end{array} \right] \left[ \begin{array}{cc} x_2 & x_3 \\ m' & m \end{array} \right] \dot{\sigma}\left(\tilde{\alpha}_m^{(L)}(x_0)\right) \dot{\sigma}\left(\tilde{\alpha}_{m'}^{(L)}(x_1)\right) \dot{\sigma}\left(\tilde{\alpha}_{m'}^{(L)}(x_2)\right) \dot{\sigma}\left(\tilde{\alpha}_m^{(L)}(x_3)\right)$$

$$\delta_{k_0 k_1} \delta_{k_2 k_3}$$

$$+n_L^{-2} \sum_{m,m'} \left[ \begin{array}{cc} x_1 & x_2 \\ m & m' \end{array} \right] \left[ \begin{array}{cc} x_3 & x_0 \\ m' & m \end{array} \right] \dot{\sigma}\left(\tilde{\alpha}_m^{(L)}(x_0)\right) \dot{\sigma}\left(\tilde{\alpha}_m^{(L)}(x_1)\right) \dot{\sigma}\left(\tilde{\alpha}_{m'}^{(L)}(x_2)\right) \dot{\sigma}\left(\tilde{\alpha}_{m'}^{(L)}(x_3)\right)$$

$$\delta_{k_0 k_3} \delta_{k_1 k_2}$$

Even though this is a very large formula one can notice that most terms are "rotation of each other". Moreover, as $n_1, ..., n_{L-1} \to \infty$, all terms containing either an $\Psi^{(L)}$, an $\Omega^{(L)}$ or a $\Gamma^{(L)}$ vanish. For the remaining terms, we may replace the NTKs $\Theta^{(L)}$ by their limit and as a result $\Psi_{k_0,k_1,k_2,k_3}^{(L+1)}(x_{i_0}, x_{i_1}, x_{i_2}, x_{i_3})$ converges to

$$n_L^{-2} \sum_m \Theta_\infty^{(L)}(x_0,x_1) \Theta_\infty^{(L)}(x_1,x_2) \Theta_\infty^{(L)}(x_2,x_3) \Theta_\infty^{(L)}(x_3,x_0) \ddot{\sigma}\left(\tilde{\alpha}_m^{(L)}(x_0)\right) \ddot{\sigma}\left(\tilde{\alpha}_m^{(L)}(x_1)\right)$$

$$\ddot{\sigma}\left(\tilde{\alpha}_m^{(L)}(x_2)\right) \ddot{\sigma}\left(\tilde{\alpha}_m^{(L)}(x_3)\right) W_{mk_0}^{(L)} W_{mk_1}^{(L)} W_{mk_2}^{(L)} W_{mk_3}^{(L)}$$

$$+n_L^{-2} \sum_m \Theta_\infty^{(L)}(x_0,x_1) \Theta_\infty^{(L)}(x_1,x_2) \Theta_\infty^{(L)}(x_2,x_3) \dot{\sigma}\left(\tilde{\alpha}_m^{(L)}(x_0)\right) \ddot{\sigma}\left(\tilde{\alpha}_m^{(L)}(x_1)\right)$$

$$\ddot{\sigma}\left(\tilde{\alpha}_m^{(L)}(x_2)\right) \dot{\sigma}\left(\tilde{\alpha}_m^{(L)}(x_3)\right) W_{mk_1}^{(L)} W_{mk_2}^{(L)} \delta_{k_0 k_3}$$

$$+n_L^{-2} \sum_m \Theta_\infty^{(L)}(x_1,x_2) \Theta_\infty^{(L)}(x_2,x_3) \Theta_\infty^{(L)}(x_3,x_0) \dot{\sigma}\left(\tilde{\alpha}_m^{(L)}(x_0)\right) \dot{\sigma}\left(\tilde{\alpha}_m^{(L)}(x_1)\right)$$

$$\ddot{\sigma}\left(\tilde{\alpha}_m^{(L)}(x_2)\right) \ddot{\sigma}\left(\tilde{\alpha}_m^{(L)}(x_3)\right) W_{mk_2}^{(L)} W_{mk_3}^{(L)} \delta_{k_0 k_1}$$

$$+n_L^{-2} \sum_m \Theta_\infty^{(L)}(x_0,x_1) \Theta_\infty^{(L)}(x_2,x_3) \Theta_\infty^{(L)}(x_3,x_0) \ddot{\sigma}\left(\tilde{\alpha}_m^{(L)}(x_0)\right) \dot{\sigma}\left(\tilde{\alpha}_m^{(L)}(x_1)\right)$$

$$\dot{\sigma}\left(\tilde{\alpha}_m^{(L)}(x_2)\right) \ddot{\sigma}\left(\tilde{\alpha}_m^{(L)}(x_3)\right) W_{mk_0}^{(L)} W_{mk_3}^{(L)} \delta_{k_1 k_2}$$

$$+n_L^{-2} \sum_m \Theta_\infty^{(L)}(x_0,x_1) \Theta_\infty^{(L)}(x_1,x_2) \Theta_\infty^{(L)}(x_3,x_0) \ddot{\sigma}\left(\tilde{\alpha}_m^{(L)}(x_0)\right) \ddot{\sigma}\left(\tilde{\alpha}_m^{(L)}(x_1)\right)$$

$$\dot{\sigma}\left(\tilde{\alpha}_m^{(L)}(x_2)\right) \dot{\sigma}\left(\tilde{\alpha}_m^{(L)}(x_3)\right) W_{mk_0}^{(L)} W_{mk_1}^{(L)} \delta_{k_2 k_3}$$

$$+ n_L^{-2} \sum_m \Theta_\infty^{(L)}(x_0, x_1) \Theta_\infty^{(L)}(x_2, x_3) \dot\sigma\left(\tilde\alpha_m^{(L)}(x_0)\right) \dot\sigma\left(\tilde\alpha_m^{(L)}(x_1)\right)$$

$$\dot\sigma\left(\tilde\alpha_m^{(L)}(x_2)\right) \dot\sigma\left(\tilde\alpha_m^{(L)}(x_3)\right) \delta_{k_0 k_1} \delta_{k_2 k_3}$$

$$+ n_L^{-2} \sum_m \Theta_\infty^{(L)}(x_1, x_2) \Theta_\infty^{(L)}(x_3, x_0) \dot\sigma\left(\tilde\alpha_m^{(L)}(x_0)\right) \dot\sigma\left(\tilde\alpha_m^{(L)}(x_1)\right)$$

$$\dot\sigma\left(\tilde\alpha_m^{(L)}(x_2)\right) \dot\sigma\left(\tilde\alpha_m^{(L)}(x_3)\right) \delta_{k_0 k_3} \delta_{k_1 k_2}$$

And all these sums vanish as $n_L \to \infty$ thanks to the prefactor $n_L^{-2}$, proving the vanishing of $\Psi_{k_0, k_1, k_2, k_3}^{(L+1)}(x_{i_0}, x_{i_1}, x_{i_2}, x_{i_3})$ in the infinite width limit.

During training, the activations $\tilde\alpha_m^{(L)}(x)$ and weights $W_{mk}^{(L)}$ move at a rate of $1/\sqrt{n_L}$ which induces a change to $\Psi^{(L+1)}$ of order $n_L^{-3/2}$ which vanishes in the infinite width limit. $\quad\square$

## D  Orthogonality of $I$ and $S$

From Lemma 2 and the vanishing of the tensor $\Gamma^{(L)}$ as proven in Lemma 2, we can easily prove the orthogonality of $I$ and $S$ of Proposition 5:

**Proposition 5.** *For any loss $C$ with BGOSS and $\sigma \in C_b^4(\mathbb{R})$, we have uniformly over $[0, T]$*

$$\lim_{n_{L-1} \to \infty} \cdots \lim_{n_1 \to \infty} \|IS\|_F = 0.$$

*As a consequence $\lim_{n_{L-1} \to \infty} \cdots \lim_{n_1 \to \infty} \mathrm{Tr}\left([I+S]^k\right) - \left[\mathrm{Tr}\left(I^k\right) + \mathrm{Tr}\left(S^k\right)\right] = 0.$*

*Proof.* The Frobenius norm of $IS$ is equal to

$$\|IS\|_F^2 = \left\| \mathcal{DYHC}(\mathcal{DY})^T (\nabla C \cdot \mathcal{HY}) \right\|_F^2$$

$$= \sum_{p_1, p_2=1}^{P} \left( \sum_{p=1}^{P} \sum_{i_1, i_2=1}^{N} \sum_{k_1, k_2=1}^{n_L} \partial_{\theta_{p_1}} f_{\theta, k_1}(x_{i_1}) c_{k_1}''(x_{i_1}) \partial_{\theta_p} f_{\theta, k_1}(x_{i_1}) \partial_{\theta_p, \theta_{p_3}}^2 f_{\theta, k_2}(x_2)(x_{i_2}) c_{k_2}'(x_{i_2}) \right)^2$$

$$= \sum_{i_1, i_2, i_1', i_2'=1}^{N} \sum_{k_1, k_2, k_1', k_2'=1}^{n_L} c_{k_1}''(x_{i_1}) c_{k_1'}''(x_{i_1'}) c_{k_2}'(x_{i_2}) c_{k_2'}'(x_{i_2'}) \Theta_{k_1, k_1'}(x_{i_1}, x_{i_1'}) \Gamma_{k_1, k_2, k_2', k_1'}(x_{i_1}, x_{i_2}, x_{i_2'}, x_{i_1'})$$

and $\Gamma$ vanishes as $n_1, ..., n_{L-1} \to \infty$ by Lemma 2.

The $k$-th moment of the sum $\mathrm{Tr}(I+S)^k$ is equal to the sum over all $\mathrm{Tr}(A_1 \cdots A_k)$ for any word $A_1 \ldots A_k$ of $A_i \in \{I, S\}$. The difference $\mathrm{Tr}\left([I+S]^k\right) - \left[\mathrm{Tr}\left(I^k\right) + \mathrm{Tr}\left(S^k\right)\right]$ is hence equal to the sum over all mixed words, i.e. words $A_1 \ldots A_k$ which contain at least one $I$ and one $S$. Such words must contain two consecutive terms $A_m A_{m+1}$ one equal to $I$ and the other equal to $S$. We can then bound the trace by

$$|\mathrm{Tr}(A_1 \cdots A_k)| \leq \|A_1\|_F \cdots \|A_{m-1}\|_F \|A_m A_{m+1}\|_F \|A_{m+2}\|_F \cdots \|A_k\|_F$$

which vanishes in the infinite width limit because $\|I\|_F$ and $\|S\|_F$ are bounded and $\|A_m A_{m+1}\|_F = \|IS\|_F$ vanishes. $\quad\square$

