# OpenReview forum: "The asymptotic spectrum of the Hessian of DNN throughout training"
_ICLR.cc/2020/Conference — Accept (Poster)_

### Official Review · AnonReviewer2 · 2019-10-22
**Official Blind Review #2**

**Rating:** 6

**Review:**

This paper discusses the behavior of the Hessian of the loss of NN during training. There is no question that the subject is relevant, and the paper gives an interesting contribution to the problem.  I do support publication.

My main criticism is in the presentation of the paper: The title and the first pages are, I believe, really misleading. Unless I am missing an important point, all derivation of the papers are being made in the so-called NTK regime. In fact, this is explicitly stated in page 3 "In the limit we study in this paper, the NTK is fixed". While this is an interesting assumption, this is by no mean a trivial one that should be stated casually in the middle of the paper. This is an extremely strong assumption (essentially, one is either in the lazy Regime of [Chizat&Bach] with infinity small learning rate, or in the kernel regime of [Jacot&al] with astronomically large layers. This is a nice setting for mathematics, but by no means a natural one and this limits drastically the conclusion of the paper: what is studied is the behavior of the DNN in the "Kernel" regime, or the "Lazy" regime. Such a limitation should be explicitly mentioned, at least in the abstract and the introduction, if not in the title itself! It should, also, be mentioned in the conclusion ("We have given an explicit formula for the limiting moments of the Hessian of DNNs throughout training").

In this respect, with the connection to the Kernel, it make sense that the Hessian is dominated by, essentially, the equivalent of Kernel PCA! This is to be expected, since one is essentially doing a Kernel Ridge Regression with random features.

Other source of confusion (for me):
* "This is contrast to the results obtained in [Pennington&Bahri] etc...." Is the only difference the fact that the network is more than two layers?
* "This gives theoretical confirmation of a number of observations about the Hessian"... I do not believe all these references are in the NTK regimes. (e.g. Chaudhari et al., for instance).

That being said, the paper is a nice and interesting contribution to the NTK regime, that is the subject of intense studies currently. I do support acceptance if the paper clarify the range of applicability of the its results.








**Experience Assessment:**

I have read many papers in this area.

**Review Assessment: Checking Correctness Of Derivations And Theory:**

I assessed the sensibility of the derivations and theory.

**Review Assessment: Checking Correctness Of Experiments:**

N/A

**Review Assessment: Thoroughness In Paper Reading:**

I read the paper at least twice and used my best judgement in assessing the paper.

---

> ### Author Response · Authors · 2019-11-09
> **Response**
>
> We agree that we should be clearer about the fact that the results during training only apply to the linear/kernel (also called lazy) regime. The framework in which we consider the neural networks will be made clearer: in particular, we will mention it in the abstract and in the introduction.
>
> Remark that our results shed new light also on the overparametrized active/mean-field training at initialization. We will add a discussion on how some of our results can be applied to the active regime at initialization. This regime is achieved by dividing the output of the network by the squared root of the width \sqrt(w). Multiplying the learning rate by w, the matrix I is not changed, whereas S is scaled by \sqrt(w). As a result, our theorems provide a full description of I and the asymptotic of the two first moments of S in the active regime (which are the same up to the new scaling). We observe that the moments of S dominate those of I. This could be a characteristic of the active regime: the "residual" matrix dominates the Fisher matrix.
>
> We regret not to have mentioned this aspect in the paper and are happy to add a discussion in the final version.
>
> * In [Pennington and Bahri] they take the second derivative of the ReLU to be zero, hence almost reducing the analysis to linear networks. One of the difference is that a non-zero second derivative leads to a non-zero first moment of S. Also, in their setting they conjecture the free independence of the two matrices I and S, whereas we can prove the asymptotic mutual orthogonality of the two matrices.
>
> * We mentioned [Chaudhari et al.] for their observations that the Hessian retains negative eigenvalues even after a very long training, in our setting this appears to be related to the existence of negative eigenvalues of S, and the fact that the rank of I is much smaller than that of S. Given our results, this phenomenon can also happen in the linear regime.
>
> We will change our wording: saying confirmation is too strong. It is more accurate to say that many empirical observations of the literature (for which it is difficult to assess in which regime they were obtained) are consistent with our theoretical results in the linear regime.

---

### Official Review · AnonReviewer3 · 2019-10-23
**Official Blind Review #3**

**Rating:** 8

**Review:**

This paper uses NTK and techniques from deriving NTK to study asymptotic spectrum of Hessian both at initialization and during training. For understanding neural network’s optimization and generalization property, understanding Hessian spectrum is quite important.
This paper gives explicit formula for limiting moments of Hessian of wide neural networks throughout training.

In detail, Hessian of neural networks can be decomposed into two components, denoted by H = I + S. In the infinite width I is totally described by NTK, and authors show that I and S are asymptotically orthogonal (both at initialization and during training).  Residual contribution is described by S, which captures evolution of Hessian by its first moments Tr (S) since Tr (S^2) remains constant and Tr (S^k ) for k>=3 vanishes.

Corollary 1 has analytic dynamics of moment of Hessian in the case of MSE loss demonstrating power of this paper’s main Theorem.  This is also supported by experiments in Figure 1.

few comments:
Authors should follow format given by ICLR style file. The paper is more dense than typical submission and may have violated page limit (10 pages max) if the style guide line was followed.
Similar to the prior comment, the reference section should be cleaned and formatted better. The reference doesn’t count towards page limit and I don’t understand the reason for them to be formatted badly and become eligible.
It would be useful if the Figure axes are more legible.
There have been many variations of NTK beyond vanilla FC networks(Arora et al. 2019, Yang 2019). Is there a major block for the analysis given in the paper to extend beyond FC networks?


**Experience Assessment:**

I have published in this field for several years.

**Review Assessment: Checking Correctness Of Derivations And Theory:**

I assessed the sensibility of the derivations and theory.

**Review Assessment: Checking Correctness Of Experiments:**

I did not assess the experiments.

**Review Assessment: Thoroughness In Paper Reading:**

I read the paper at least twice and used my best judgement in assessing the paper.

---

> ### Author Response · Authors · 2019-11-09
> **Response**
>
> Sorry for the formating, we have fixed these issues and the article fits well within the 10 pages limit with the right formating. We will improve the formating of the bibliography and the figures.
>
> Regarding the generalizations to other architectures, we are confident that the proofs can be extended. Indeed the proofs of convergence of the kernels in this article and of the NTK are closely related and, just like for the NTK, they could be generalized to convolutional networks. However doing so would greatly increase the complexity of the proofs which already involve heavy notations due to the introduction of important new kernels and tensors. We intend to carry out a more systematic study of this family of tensors: the generalization to other architectures would be clearer in this broader setting.

---

### Official Review · AnonReviewer1 · 2019-10-27
**Official Blind Review #1**

**Rating:** 3

**Review:**

This paper uses the Neural Tangent Kernel (NTK) to presents an asymptotic analysis of the evolution of Hessian of the loss (w.r.t. model parameters) throughout training. The authors leverage the Neural Tangent Kernel to analyze the evolution of the Hessian of the loss w.r.t the model parameters. Specifically the authors show that as the width of the neural networks tend to infinity, the Hessian can be decomposed into two components: (1) one that reflects the initialization of the model parameters and reduces as training progresses; and (2) one that captures the principal directions of the data.

Technical Soundness: The paper appears to be technically sound, though I did not go through the 14 pages of proofs.

Potential Impact:
I found the focus of the paper to be quite narrow which I think will negatively affect the impact that this paper could have. The authors take the reader on a notational taxing voyage through the decomposition of the Hessian to finally arrive at a result that is rather intuitive if not entirely obvious. While I learned something about the evolution of the Hessian (in the limit of the infinitely wide NN),  I can not say that this paper will have significant impact on how I think about NN training.

To address this issue of significance and impact: What novel conclusions about Neural Network learning dynamics can you draw from your analysis? What are the implications for generalization or for future training algorithms?

Clarity: Beyond the possibly unduly heavy notation, the paper is rather clear and well written. There is a minor typo in the first equation of Sec. 2.3 (i -> j).


**Experience Assessment:**

I have read many papers in this area.

**Review Assessment: Checking Correctness Of Derivations And Theory:**

I assessed the sensibility of the derivations and theory.

**Review Assessment: Checking Correctness Of Experiments:**

I carefully checked the experiments.

**Review Assessment: Thoroughness In Paper Reading:**

I read the paper thoroughly.

---

> ### Author Response · Authors · 2019-11-09
> **Response**
>
> Let us first emphasize that while our work is primarily motivated by the desire to give a clean understanding of the Hessian of DNNs (a subject that has seen a lot of empirical and theoretical research activity), it has concrete implications for the understanding of the training of DNNs. The introduction of the NTK has indeed shown that the dynamics of DNNs is asymptotically approximated by their linearizations around initialization; if we were to follow that picture naively, however, we would get that the matrix S is zero, which we show is not the case. So, while we keep *an exact description*, we are able to show a *nontrivial difference between very wide DNNs and their linear (kernelized) approximations* during the training.
>
> This difference can also be used to understand the cross-over between the linear/kernel regime and the so-called "active" regime (this is discussed in response to reviewer #2): we observe that in the linear regime the Fisher matrix I dominates the matrix S, while in the active regime, S becomes the dominant matrix (at initialization). Connecting our results with the empirical works on the Hessian is hence a way to determine what is the regime in which they are taking place.
>
> The analysis moreover leads us to the following insights:
>
> 1. The vanishing of the residual S in operator norm but not in Frobenius norm at initialization.
>
> 2. The spectrum of S is not exactly symmetric because its first moment does not vanish in contrast to the observations of [Pennington and Bahri].
>
> 3. The dominating eigenvalues in the Hessian leading to a narrow valleys in the loss appear in the ordered/freeze regime as constant modes [Papyan].

---

### Decision · Program_Chairs · 2019-12-19

**Decision:**

Accept (Poster)

**Comment:**

This paper studies the spectrum of the Hessian through training, making connections with the NTK limit. While many of the results are perhaps unsurprising, and more empirically driven, together the paper represents a valuable contribution towards our understanding of generalization in deep learning. Please carefully account for the reviewer comments in the final version.